# Altered regulation of mesenchymal cell senescence in adipose tissue promotes pathological changes associated with diabetic wound healing

Arisa Kita [1], Yuki Saito [2✉], Norihiro Miura[3], Maki Miyajima[3], Sena Yamamoto[3], Tsukasa Sato[3], Takatoshi Yotsuyanagi[1], Mineko Fujimiya[2] & Takako S. Chikenji [2,3✉]

Pathologic diabetic wound healing is caused by sequential and progressive deterioration of hemostasis, inflammation, proliferation, and resolution/remodeling. Cellular senescence promotes wound healing; however, diabetic wounds exhibit low levels of senescent factors and accumulate senescent cells, which impair the healing process. Here we show that the number of p15$^{INK4B}$ + PDGFRα + senescent mesenchymal cells in adipose tissue increases transiently during early phases of wound healing in both non-diabetic mice and humans. Transplantation of adipose tissue from diabetic mice into non-diabetic mice results in impaired wound healing and an altered cellular senescence–associated secretory phenotype (SASP), suggesting that insufficient induction of adipose tissue senescence after injury is a pathological mechanism of diabetic wound healing. These results provide insight into how regulation of senescence in adipose tissue contributes to wound healing and could constitute a basis for developing therapeutic treatment for wound healing impairment in diabetes.

[1] Department of Plastic and Reconstructive Surgery, Sapporo Medical University, Sapporo, Japan. [2] Department of Anatomy, Sapporo Medical University School of Medicine, Sapporo, Japan. [3] Graduate School of Health Sciences, Hokkaido University, Sapporo, Japan. ✉email: yuki.saito@sapmed.ac.jp; chikenji@pop.med.hokudai.ac.jp

Cellular senescence is a process in which cells cease dividing, and it is activated in response to numerous stressors, including exposure to genotoxic agents, nutrient deprivation, hypoxia, mitochondrial dysfunction, and oncogene activation[1]. One of the features of senescent cells is that they undergo distinctive phenotypic alterations called the senescence-associated secretory phenotype (SASP), wherein they secrete cytokines, chemokines, growth factors, and matrix metalloproteases (MMPs)[1–3]. Senescent cells exert beneficial or detrimental effects on both tissue remodeling and dysfunction through cell-to-cell communication and SASP expression[2,3]. Beneficial effects of senescence include tissue remodeling and regeneration. Increased senescent cells are observed in mouse skin during normal wound healing, heart regeneration in zebrafish and neonatal mice, and tissue regeneration in salamanders and zebrafish[4–8]. By contrast, chronically accumulated senescent cells exhibit an anti-apoptotic phenotype and a SASP, which induce detrimental effects including lung fibrosis, kidney and heart dysfunction, impairment of the central neuronal system, and muscle weakness[9].

During normal wound healing, the numbers of senescent fibroblasts and endothelial cells increase transiently in the skin, and wound closure is accelerated by the secretion of platelet-derived growth factor-AA (PDGF-AA), which induces myofibroblast differentiation[8]. Senescence is also induced in fibroblasts during skin wound healing by either cellular communication network factor 1 (CCN1) or CCN2, which both inhibit excess fibrosis[10,11]. Transiently induced to allow matrix formation and then regresses. By contrast, chronic diabetic wounds show chronically larger numbers of senescent fibroblasts and macrophages that produce high levels of matrix-degrading proteases and inflammatory cytokines. These enzymes cause uncontrolled inflammation and inhibit wound healing[12–15]. When more than 15% of cells isolated from venous ulcers are senescent fibroblasts, wound healing is delayed[16]. However, the regulatory mechanisms that govern these properties of cell senescence, specifically concerning their contribution to modifying tissue homeostasis and pathological conditions by producing various cytokines and growth factors, remain unknown.

Adipose tissue is attracting attention for promoting wound healing[17–21]. Within subcutaneous adipose tissue, stromal vascular cells and their subsets release growth factors and cytokines critical for neovascularization and wound repair[21]. Subcutaneous tissue has garnered attention particularly in obesity and type-2 diabetes because increased inflammation can alter the outcome of wound repair through the production of large numbers of hormones and cytokines, such as tumor necrosis factor α (TNF-α), Interleukin 6 (IL-6), and plasminogen activator inhibitor 1 (PAI-1)[22]; these proteins are the major components of the SASP. Senescent cell accumulation in the adipose tissue of patients with diabetes and obesity is associated with insulin resistance and systemic inflammation[23–27]. Furthermore, subcutaneous adipose tissue in obese subjects is characterized by an excessive amount of interstitial fibrosis and phenotypic changes in human pre-adipocytes, which may contribute to tissue deterioration[28].

In this study, we investigated cellular senescence in skin and subcutaneous adipose tissue during the wound healing process using a $Lepr^{db/db}$ type-2 diabetic mouse model. We found that growth of subcutaneous adipose tissue reached the wound site in control mice; however, in the diabetic $Lepr^{db/db}$ mice, the adipose tissue remained below the granulating tissue, and thereby impaired wound healing. In addition, transplantation of $Lepr^{db/db}$-derived adipose tissue into control mice impaired wound healing. In the adipose tissue of control mice, senescent cells increased at 2 days post-wound (DPW) and decreased at 8 DPW, whereas senescent cells in $Lepr^{db/db}$ mice remained low at 2 DPW and

increased slightly at 8 DPW. The composition of SASP factors differs between the control and $Lepr^{db/db}$ mice, resulting in the inhibition of fibroblast migration. Our results demonstrate that diabetic adipose tissue impairs transient senescence during pathological healing, which causes deteriorated wound healing, suggesting that cellular senesces of adipose tissue could be a therapeutic target for diabetic ulcers.

## Results

### Type-2 diabetic mice models have impaired wound healing.
C57BLKS/J Iar -+ Lepr$^{db}$/+Lepr$^{db}$ ($Lepr^{db/db}$) and C57BLKS/J Iar -m + /+Lepr$^{db}$ ($Lepr^{db/+}$) mice were used in this study as type-2 diabetic and control mouse models, respectively. Blood glucose level ≤200 mg/dL ($Lepr^{db/+}$) or > 300 mg/dL (type-2 diabetic, $Lepr^{db/db}$) was used to confirm the absence or presence of diabetes, respectively (Fig. 1a). Full-thickness excisional skin wounds were created on the backs of $Lepr^{db/db}$ and $Lepr^{db/+}$ mice, and the wound closure rate was evaluated at 2, 5, and 8 DPW. At each time-point, the wound closure rates were impaired to a greater extent in $Lepr^{db/db}$ mice than in $Lepr^{db/+}$ mice (Fig. 1b–c). Histological analysis involving hematoxylin and eosin (H&E) staining and Masson trichrome staining, indicated the presence of adipose tissues immediately under the wound eschar region in $Lepr^{db/+}$ mice at 2 DPW albeit not in $Lepr^{db/db}$ mice (Fig. 1d). At 8 DPW in $Lepr^{db/+}$ mice, there was less adipose tissue under the wound eschar as the wound closed. Conversely, in $Lepr^{db/db}$ mice, we observed increased cell infiltration beneath the wound eschar region, and thus more adipose tissue and collagen deposition at the wound closing region (Fig. 1d). Our quantitative data indicates significantly more cell infiltration and collagen deposition in $Lepr^{db/db}$ mice along with significantly less area covered by adipose tissue (Fig. 1e–f).

Next, we used antibodies specific for platelet-derived growth factor receptor α (PDGFRα) and α smooth muscle actin (α-SMA) to identify mesenchymal stromal cells that produce ECM and play important roles in wound healing of subcutaneous tissue of the skin[29]. PDGFRα is a cell surface receptor tyrosine kinase expressed in mesenchymal stromal cells in a variety of tissues and is also used as a marker for adipose progenitor cells[30–33]. Differentiation of mesenchymal stromal cells into α-SMA + myofibroblasts is necessary for acute wounds to heal but increased PDGFRα activation causes progressive fibrosis in many organs, including the skin[8,34]. Immunohistochemical analysis indicated a greater percentage of α-SMA + cells in interstitial connective tissue in $Lepr^{db/+}$mice (Fig. 1h–i, and Supplementary Fig. 1). However, $Lepr^{db/db}$ mice showed a higher percentage of PDGFR-α + cells but a lower percentage of α-SMA + cells in interstitial connective tissue compared with $Lepr^{db/+}$mice (Fig. 1h–i). In addition, $Lepr^{db/db}$ mice showed an increased population of cells positive for both PDGFRα and α-SMA in interstitial connective tissue (Fig. 1h–i).

The panniculus carnosus is a thin striated muscular layer with interstitial connective tissue under it[35]. In mouse, interstitial connective tissue separates the panniculus carnosus from subcutaneous adipose tissue. We concluded that $Lepr^{db/db}$ mice exhibit diffuse distribution of α-SMA and PDGFRα + cells within interstitial connective tissue and granulation tissue at 8 DPW after observing a lower percentage of α-SMA + cells along with a higher percentage of cells positive for both PDGFRα and α-SMA in their granulation tissue (Fig. 1h, j).

### Transplantating adipose tissue-derived from Lepr$^{db/db}$ mice into Lepr$^{db/+}$ mice impairs wound healing. To elucidate the role that adipose tissue plays in wound healing impairment, adipose tissue derived from $Lepr^{db/db}$ mice was transplanted into the excisional

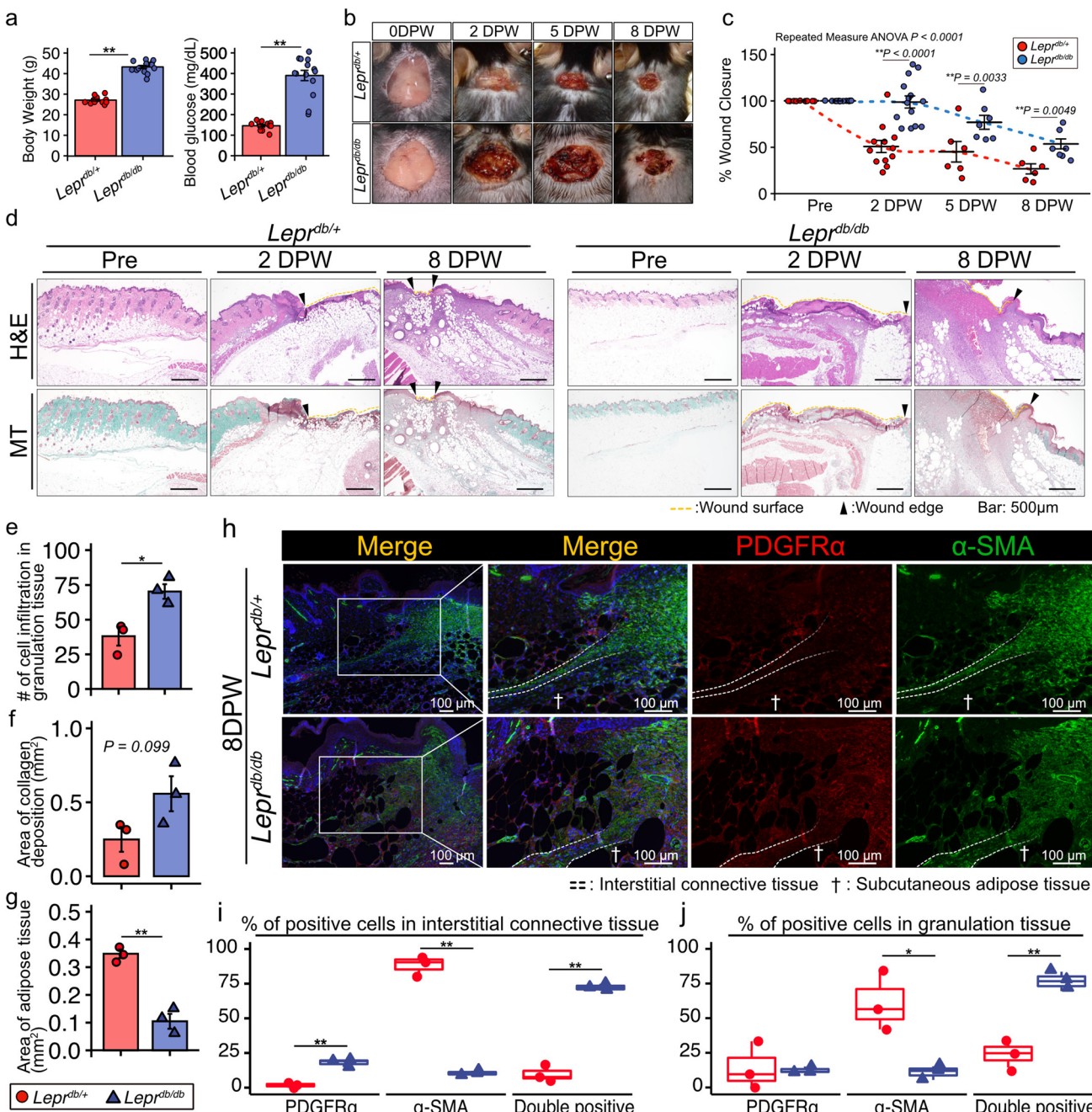

**Fig. 1 *Lepr^{db/db}* mice exhibit impaired wound healing. a** Body weight and blood glucose levels for *Lepr^{db/+}* and *Lper^{db/db}* mice before full-thickness excisional skin wounds were created ($n = 12$ for *Lepr^{db/+}* and $n = 14$ for *Lper^{db/db}* mice). **b** Representative wound images during the healing process and (**c**) the wound area, reported as the percent of wound closure compared to the 0 days post wound (0 DPW) area ($n = 12$ for *Lepr^{db/+}* 0 and 2 DPW, $n = 6$ for *Lepr^{db/+}* 5 and 8 DPW, $n = 14$ for *Lper^{db/db}* 0 and 2 DPW, and $n = 7$ for *Lepr^{db/db}* 5 and 8 DPW). **d** Representative images of H&E staining and Masson trichrome (MT) staining of a wound at pre-wounding, 2 DPW, and 8 DPW. **e** Quantitative data concerning the number of infiltrated cells. **f** Area covered by collagen deposits. **g** Area covered by adipose tissue ($n = 3$ for each group). **h–j** Representative images of a wound at 8 DPW following immunostaining for PDGFRα and α-SMA (**h**), and percentage of cells positive for either PDGFRα, α-SMA, or both in interstitial connective tissue (**i**) and in granulation tissue (**j**) ($n = 3$ for each group). Quantitative data are shown as means ± SE in dot plots or box-and-whisker plots with IQRs and 1.5 times the IQR. *p*-values were determined using either a two-tailed Student's *t*-test or the Tukey method for one-way ANOVA (*$p < 0.05$ and **$p < 0.001$).

skin wound region of *Lepr^{db/+}* mice. The excisional skin wound was sutured closed after transplantation to stabilize the transplanted tissue (Fig. 2a). We evaluated the histological wound healing score at 8 DPW using H&E staining and Masson trichrome staining[36]. Three investigators each assigned a histological score in a blinded fashion, and the corresponding intraclass correlation coefficient (ICC) was 0.79. The histological wound healing scores for mice

transplanted with *Lepr^{db/db}* adipose tissue (*Lepr^{db/db}* ATT mice) were significantly lower than those for mice transplanted with *Lepr^{db/+}* adipose tissue (*Lepr^{db/+}* ATT mice) and for sham control mice (Fig. 2b–c). Immunohistochemical analysis indicated a higher percentage of PDGFRα + cells and a lower percentage of α-SMA + cells in the interstitial connective tissue of *Lepr^{db/db}* ATT mice compared with *Lepr^{db/+}*ATT mice (Fig. 2d–e). Moreover, cells

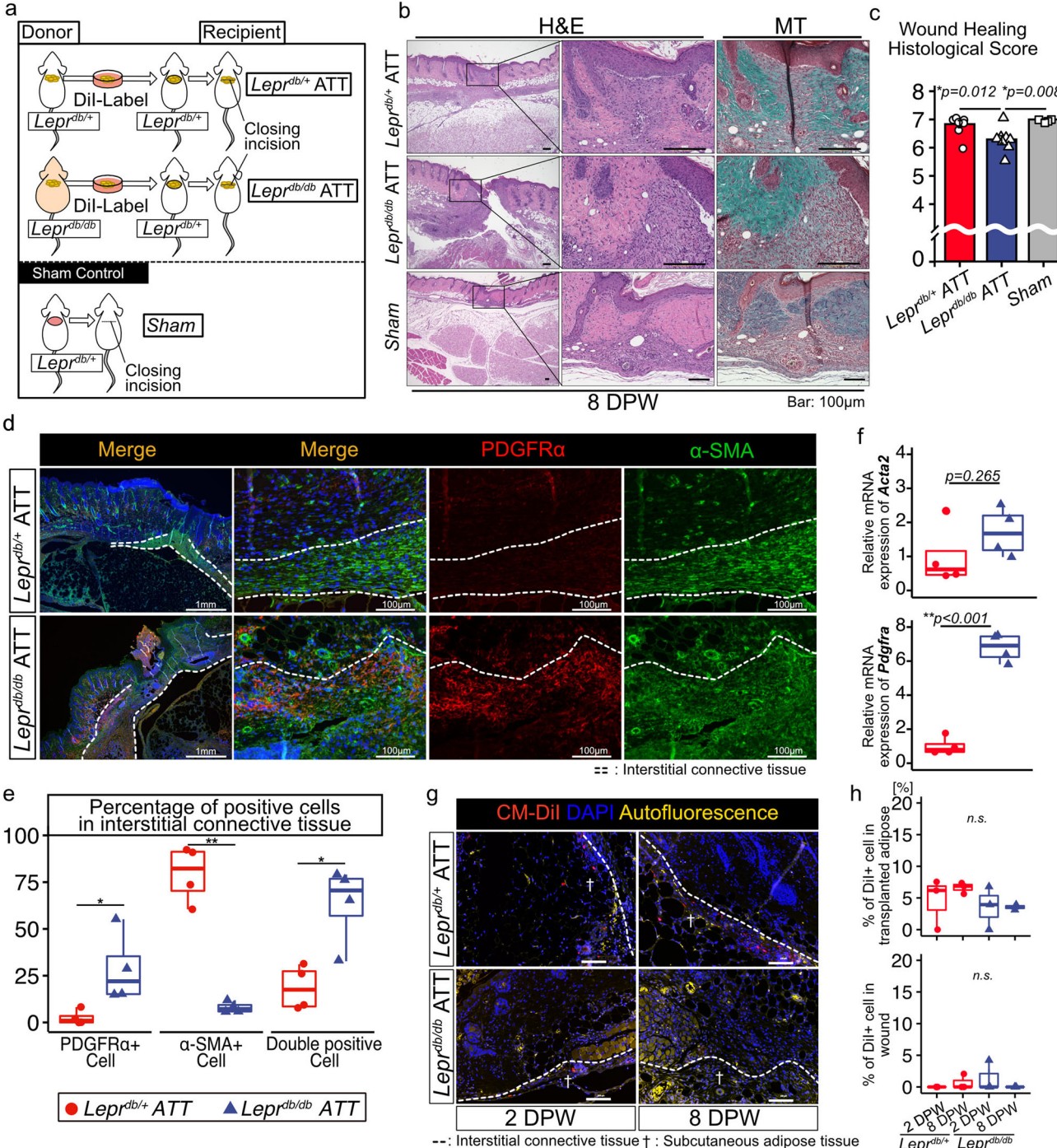

**Fig. 2 Transplantation of adipose tissue derived from *Lepr^db/db* mice into *Lepr^db/+* mice impairs wound healing. a** Schematic diagram of the procedure for the adipose tissue transplantation (ATT) experiments. **b** Representative image of H&E staining and MT staining of a wound at 8 DPW and **(c)** the quantitative analysis of the wound healing histological score (*n* = 8 in the ATT group and *n* = 4 in the sham group). **d, e** Representative images of PDGFRα and α-SMA immunostaining of a wound edge at 8 DPW and quantitative data on the percentage of PDGFRα and α-SMA + cells in interstitial connective tissue (*n* = 4 for each group). **f** Relative levels of *Acta2* and *Pdgfra* mRNA in *Lepr^db/+* and *Lepr^db/db* mouse skin at 8 DPW. **g, h** Localization of CM-DiI-labeled adipose tissue–derived cells within wound regions and quantitative data (*n* = 3 for each group). Quantitative data are shown as means ± SE in either dot plots or box-and-whisker plots with IQRs and 1.5 times the IQR. *p*-values were determined using a two-tailed Student's *t*-test or the Tukey method for one-way ANOVA (**p* < 0.05 and ***p* < 0.001).

positive for both PDGFRα and α-SMA were more prevalent in the interstitial connective tissue of *Lepr^db/db* ATT mice (Fig. 2d–e), which was also observed for impaired wound healing in *Lepr^db/db* mice. We also detected greater expression of *Pdgfra* but not *Acta2* in skin cells of *Lepr^db/db* ATT mice using quantitative reverse transcription PCR (qRT-PCR) (Fig. 2f). These results suggest that

transplantation of adipose tissue derived from *Lepr^db/db* mice into *Lepr^db/+* mice impairs wound healing.

To ascertain the role of cell movement in transplanted adipose tissue, we investigated the localization of CM-DiI–labeled adipose tissue–derived cells in wound regions. We found CM-DiI+ adipose cells in layers below the interstitial connective tissue at 2

DPW and 8 DPW in both $Lepr^{db/+}$ ATT and $Lepr^{db/db}$ ATT mice. We also observed that CM-DiI+ cells remained under interstitial connective tissue over time (Fig. 2g, h).

**Senescence-related gene expression levels were transiently increased in subcutaneous adipose tissue during wound healing.** Senescent cells play beneficial roles in wound healing by expressing SASP factors including PDGF-AA, CCN1, VEGF, and Serpine1, which promote the production of ECM and prevent excessive fibrosis[8]. By contrast, accumulation of senescent cells is observed in diabetic ulcers[15]. We speculated that different types of cellular senescence occur between normal and diabetic wound healing in response to wounds. Hence, we investigated the time-dependent change in the expression of senescence-related factors during wound healing in $Lepr^{db/db}$ and $Lepr^{db/+}$ mice. We harvested the skin and subcutaneous adipose tissue at pre-wounding, 2 DPW, and 8 DPW. Mouse skin and adipose tissue are divided by interstitial connective tissue.

In skin tissue, damage induced the expression of cellular senescence markers. $Cdkn1a$ mRNA transcription levels increased at 8 DPW relative to those at pre-wound and 2 DPW in both $Lepr^{db/db}$ and $Lepr^{db/+}$ mice (Fig. 3a). $Trp53$ transcript level decreased at 2 DPW relative to that of the pre-wound level in $Lepr^{db/+}$ mice, but the level in $Lepr^{db/db}$ mice was not significantly different (Fig. 3a). $Cdkn2b$ is a marker of cell senescence[37,38]. The transcription level of $Cdkn2b$ mRNA increased at 2 DPW relative to that of the pre-wound level in $Lepr^{db/+}$ mice and decreased at 8 DPW relative to that of the level at 2 DPW in $Lepr^{db/db}$ mice (Fig. 3a). The mRNA transcription level of the SASP-related factor $Serpine1$ was essentially the same in both mouse models, but the $Serpine2$ transcription level increased at 8 DPW in $Lepr^{db/db}$ mice (Fig. 3a).

Next, we measured mRNA levels in subcutaneous adipose tissue during wound healing. The levels of $Cdkn2b$ and $Trp53$ mRNA in $Lepr^{db/+}$ mice were higher at 2 DPW than at pre-wounding while the level of $Cdkn2b$ transcript was lower at 8 DPW (Fig. 3a). In $Lepr^{db/db}$ mice, $Cdkn2b$ and $Trp53$ transcription levels did not significantly change during wound repair, and increased levels of $Cdkn1a$, $Serpine2$, and $Tgfb1$ transcription, relative to 2 DPW, were observed at 8 DPW (Fig. 3a). The mRNA transcription levels of $Serpine1$ and $Il6$ did not change in $Lepr^{db/+}$ mice during wound healing, but $Lepr^{db/db}$ mice exhibited a time-dependent decrease in $Serpin1$ and a transient increase in $Il6$ levels (Fig. 3a). To clarify these differences in gene expression patterns related to senescence during the wound healing process between $Lepr^{db/+}$ and $Lepr^{db/db}$, principal component analysis (PCA) was performed (Fig. 3b–c). In skin tissue, PCA plots indicate that both $Lepr^{db/+}$ mice and $Lepr^{db/db}$ mice had similar patterns related to senescence during the wound healing process (Fig. 3b). Conversely, PCA plots in adipose tissue indicate that $Lepr^{db/+}$ mice experienced large changes in senescence-related gene transcription at 2 DPW, whereas at 8 DPW, the transcription of these genes returned to similar levels to what was observed pre-wounding (Fig. 3c). By contrast, $Lepr^{db/db}$ mice showed minimal changes in senescence-related gene transcription during wound healing (Fig. 3c).

To identify the cell type that exhibits senescence in subcutaneous adipose tissue, we performed immunohistochemistry for p15INK4B and PDGFRα. Encoded by $CDKN2B$, p15INK4B is an INK4 class of cell-cycle inhibitors[37,39]. We verified that the percentage of PDGFRα + cells was higher at 8 DPW in $Lepr^{db/db}$ adipose tissue and that $Pdgfra$ mRNA expression was higher in 8 DPW but significantly lower in $Lepr^{db/+}$ adipose tissue (Fig. 4a–c). In $Lepr^{db/+}$ adipose tissue, the percentage of p15INK4B + in PDGFRα + cells increased at both 2 and 8 DPW, and the

fluorescence intensity of p15INK4B in PDGFRα + cells transiently increased at 2 DPW (Fig. 4a, d–e). However, in $Lepr^{db/db}$ adipose tissue, an increase in the percentage of p15INK4B + in PDGFRα + cells was delayed at 8 DPW relative to that observed for $Lepr^{db/+}$ mice (Fig. 4a, d–e). In addition, p15INK4B fluorescence intensity in PDGFRα + cells in $Lepr^{db/db}$ mice was higher at 2 DPW and was similarly high at 8 DPW (Fig. 4a, d–e). To determine if p15INK4B is a senescence marker, we performed co-immunostaining with anti-p15INK4B, p16INK4A, and p21CIP1/WAF1 antibodies. We observed that 75% of p15INK4B cells expressed either p16INK4A, p21CIP1/WAF1, or both (Supplementary Fig. 2a–b).

We also determined the expression of SPiDER-β-Gal, a fluorescent probe for detecting SA-β-Gal, in PDGFR-α + cells. In $Lepr^{db/+}$ adipose tissue, the percentage of SPiDER-β-Gal+ in PDGFRα + cells was higher at 2 DPW and lower at 8 DPW (Supplementary Fig. 3a–b). However, in $Lepr^{db/db}$ adipose tissue, the percentage of SPiDER-β-Gal+ in PDGFRα + cells was higher at 8 DPW than at pre-wounding and 2 DPW (Supplementary Fig. 3a–b).

These results suggest that cellular senescence rapidly occurs in $Lepr^{db/+}$ adipose tissue after wounding, and cell senescence is transient. However, in $Lepr^{db/db}$ adipose tissue, cell senescence is delayed after wounding, and this delay may allow for accumulation of senescent cells in diabetic wound healing.

**Senescent PDGFRα + cells accumulate in subcutaneous adipose tissue during wound healing in diabetic patients.** To determine whether the number of senescent cells increases in subcutaneous adipose tissue during wound healing in diabetic and non-diabetic patients, we performed immunohistochemical analysis of PDGFRα + cells and looked for expression of p15INK4B and γH2A.X, which is a DNA damage–induced cell senescence marker[40]. Patient demographics are presented in Table 1. In non-diabetic patients, a negative correlation was observed between the percentage of p15INK4B+ cells in PDGFRα + cells and the time post-wounding (Fig. 4a, c). By contrast, in diabetic patients, we observed a high positive correlation between the percentage of PDGFRα + cells that were either p15INK4B or γH2A.X + and the time post-wounding (Fig. 4b, c, d). To determine if p15INK4B is a marker of senescence in human adipose tissue, we performed co-immunostaining with antibodies against p15INK4B, p16INK4A, and p21CIP1/WAF1. We found that 80% of cells that express p15INK4B also express either p16INK4A, p21CIP1/WAF1, or both (Supplementary Fig. 4a–c). These results indicate that cell senescence gradually accumulates in diabetic adipose tissue during wound healing, but in non-diabetic patients, cell senescence occurs robustly in adipose tissue during the early stages of wound healing.

**SASP factors derived from Lepr$^{db/db}$ mice adipose tissue impair wound healing.** Overall, our results suggest that the difference in cell senescence between diabetic and non-diabetic adipose tissue influences wound healing. Because adipose-derived cells stay under the interstitial connective tissue during wound healing, SASP factors may contribute to wound healing. Therefore, we determined the differences in the expression levels of SASP factors in $Lepr^{db/+}$ and $Lepr^{db/db}$ adipose tissue. Adipose tissue was collected from $Lepr^{db/+}$ and $Lepr^{db/db}$ mice at pre-wound, 2 DPW, and 8 DPW, and organ culture was performed to harvest SASP-containing culture media. A proteome profiler antibody array was used to investigate whether SASP factors were present in the culture media. At pre-wound, adipose tissue in $Lepr^{db/+}$ mice had higher expression levels of IGFBP3 and IGFBP5, but $Lepr^{db/db}$ mice have higher expression levels of CCL6 and CCL11. At 2 DPW, both $Lepr^{db/+}$ and $Lepr^{db/db}$

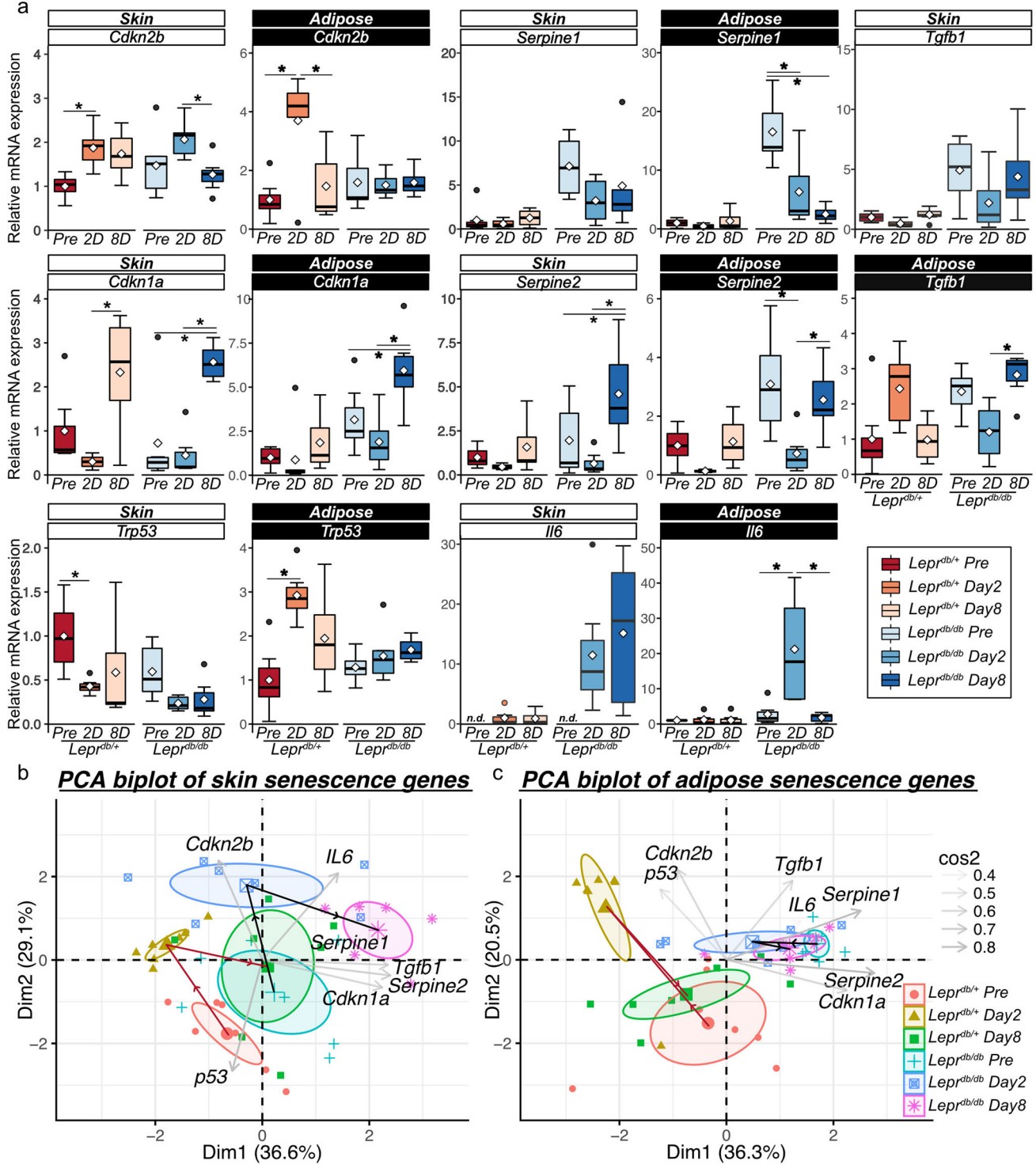

**Fig. 3 Cellular senescence in subcutaneous adipose tissue during wound healing in *Lepr^db/+* and *Lepr^db/db* mice. a** Relative mRNA expression of senescence-related genes at pre-wound, 2 DPW, and 8 DPW in *Lepr^db/+ and Lepr^db/db* mice (*n* = 7 for each group). **b, c** Principal component analysis (PCA) of the levels of senescence based on senescence-related gene expression at pre-wound, 2 DPW, and 8 DPW for *Lepr^db/+ and Lepr^db/db* mice in skin and adipose tissue. Quantitative data are presented as means and medians with IQRs and 1.5 times the IQR and are shown as box-and-whisker plots. *p*-values were determined using the Tukey method for one-way ANOVA (**p* < 0.05).

adipose tissue had increased expression levels of various secretory factors; however, the composition was different (Fig. 5a, b, and Supplementary Fig. 5). The *Lepr^db/+* adipose tissue exhibited higher expression levels of Adiponectin, Ang2, CCL2, CRP, CXCL3, IL1Ra, MMP9, VEGF, and CCN4 (Fig. 5a, b, and Supplementary Fig. 5). By contrast, *Lepr^db/db* adipose tissue had higher

expression levels of IL11, CCL11, and MMP3 (Fig. 5a, b, and Supplementary Fig. 5). At 8 DPW, the levels of VEGF, IGFBP family members, and Serpin F1, which are important factors for wound healing[8,41], increased in *Lepr^db/+* adipose tissue–derived conditioned media, but *Lepr^db/db* conditioned media continued to have increased expression levels of CCL6, CCL11, and CXCL2

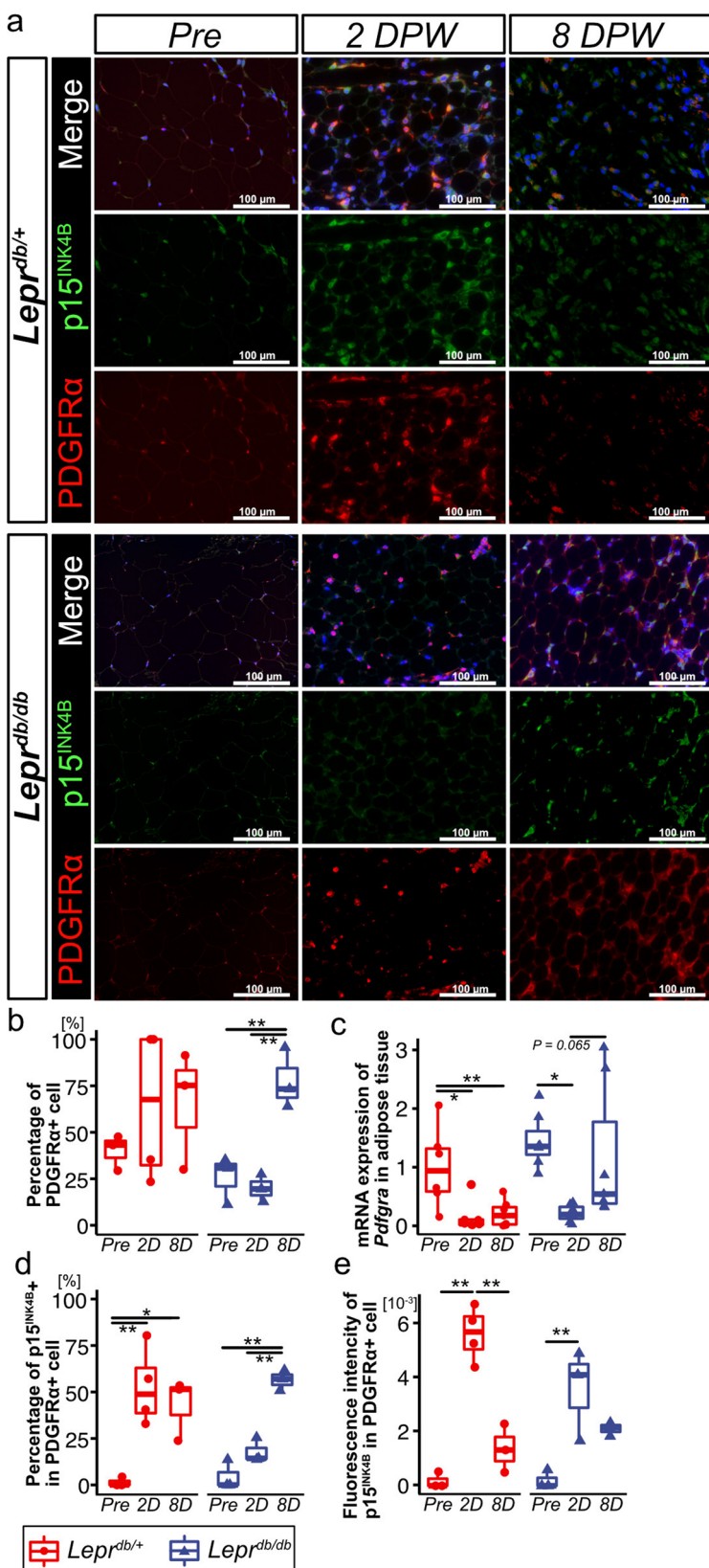

(Fig. 5a, b, and Supplementary Fig. 5). Finally, to determine whether the conditioned media derived from $Lepr^{db/+}$ or $Lepr^{db/db}$ adipose tissue promotes or inhibits wound healing, we performed wound healing assays that involved treating wounds with conditioned media collected from 2DPW adipose tissue. We found that the conditioned medium derived from $Lepr^{db/+}$ adipose tissue promotes wound closure relative to the control medium at 1 and 2 d after wounding, while the conditioned medium derived from $Lepr^{db/db}$ adipose tissue delays wound closure relative to $Lepr^{db/+}$ adipose tissue at 3 and 4 d after wounding (Fig. 6c, d).

**Fig. 4 p15$^{INK4B}$ expression in PDGFRα + cells in subcutaneous adipose tissue during wound healing in *Lepr$^{db/+}$* and *Lepr$^{db/db}$* mice. a** Representative images of adipose tissue following immunostaining for PDGFRα and p15$^{INK4B}$. Samples were collected at pre-wounding, 2 DPW, and 8 DPW. **b, c** Percentage of PDGFRα + cells (*Lepr$^{db/+}$*pre-wounding: n = 3; *Lepr$^{db/+}$* 2 DPW: n = 4; *Lepr$^{db/+}$* 8 DPW; n = 3; *Lepr$^{db/db}$* pre-wounding: n = 3; *Lepr$^{db/db}$* 2 DPW: n = 3; *Lepr$^{db/db}$* 8 DPW; n = 3) and *Pdgfra* mRNA levels in adipose tissue at pre-wounding, 2 DPW, and 8 DPW (*Lepr$^{db/+}$*pre-wounding: n = 6; *Lepr$^{db/+}$* 2 DPW: n = 7; *Lepr$^{db/+}$* 8 DPW; n = 7; *Lepr$^{db/db}$* pre-wounding: n = 7; *Lepr$^{db/db}$* 2 DPW: n = 7; *Lepr$^{db/db}$* 8 DPW; n = 6). **d, e** Percentage of PDGFRα − and p15$^{INK4B}$ + cells and fluorescence intensity of p15$^{INK4B}$ in PDGFRα + cells (*Lepr$^{db/+}$*pre-wounding: n = 3; *Lepr$^{db/+}$* 2 DPW: n = 4; *Lepr$^{db/+}$* 8 DPW; n = 3; *Lepr$^{db/db}$* pre-wounding: n = 3; *Lepr$^{db/db}$* 2 DPW: n = 3; *Lepr$^{db/db}$* 8 DPW; n = 3). Quantitative data are presented as box-and-whisker plots with IQRs and 1.5 times the IQR. *p*-values were determined using the Tukey method for one-way ANOVA (*$p < 0.05$ and **$p < 0.001$).

## Discussion

Cellular senescence contributes to wound healing in both the normal and pathological healing processes[8,10–14,16,42]. Therefore, it is critical to understand the role that cellular senescence plays in wound healing impairment, and modulating this senescence presents a therapeutic approach to reduce this impairment. In normal wound healing, the number of senescent cells in mice skin increases 3–6 DPW and decreases 9 DPW, and genetic depletion of p16$^{INK4A}$-expressing senescent cells delays wound healing[8]. Senescent fibroblasts and endothelial cells in the dermis accelerate wound closure by secreting platelet-derived growth factor (PDGF) AA[8], which binds to its tyrosine kinase receptor PDGFα, thereby inducing myofibroblast differentiation[43,44].

Given that the expression pattern of senescence-related genes, including *Cdkn2b*, by PCA did not differ between *Lepr$^{db/db}$* and *Lepr$^{db/+}$* in skin cells post-wounding, skin senescence in diabetic wounds should be similar to that during normal wound healing. In contrast to expectations, we observed a rapid but transient increase in senescent cells in subcutaneous adipose tissue for the control; however, the number of senescent cells in diabetic *Lepr$^{db/db}$* adipose tissue gradually increased post-wounding. Transplantation of subcutaneous adipose tissue of diabetic *Lepr$^{db/+}$* into normal dermal tissue inhibits wound healing and leads to more PDGFRα + cells in *Lepr$^{db/db}$* interstitial connective tissue layer, while α-SMA + myofibroblasts of interstitial connective tissue infiltrated the wound site with adipose tissue in control. These data indicate that the senescence function of subcutaneous adipose tissue may induce PDGFRα + cells to differentiate into myofibroblasts in the control interstitial connective tissue layer, but the delay in cellular senescence and the difference in SASP components in diabetic subcutaneous adipose tissue may limit the differentiation of PDGFRα + cells into myofibroblasts, resulting in impaired wound healing.

We found that SASP factor expression levels differ between *Lepr$^{db/db}$*- and diabetic *Lepr$^{db/db}$*-derived adipose tissue, which affects wound healing in vitro. In diabetic wounds, VEGF and CCL2 enhance wound healing by promoting angiogenesis[21] and regulating macrophages[45], respectively. VEGF secretion increased at 2 DPW in *Lepr$^{db/+}$* mouse adipose tissue but not in *Lepr$^{db/db}$* adipose tissue. CCL2 secretion also increased at 2 DPW in adipose tissue in *Lepr$^{db/+}$*mice. The levels of CCN4, a senescence regulator[46] that enhances wound healing by regulating dermal fibroblast cell migration, proliferation, and ECM expression[47], increased at 2 DPW in *Lepr$^{db/+}$* mouse adipose tissue. By contrast, diabetic *Lepr$^{db/db}$* adipose tissue exhibited higher secretion of IL11, CCL11, and MMP3 at 2 DPW. IL11 is a member of the IL6 family of cytokines and binds to IL11 receptor subunit alpha (IL11RA), which is expressed on stromal cells and promotes tissue fibrosis[48,49]. In atopic dermatitis, the number of IL11-expressing cells is elevated in skin biopsy specimens relative to controls, and a significant correlation exists between IL11 and type I collagen deposition[50].

MMPs, known SASP factors[1], are important regulators of ECM degradation and deposition, and the timing and level of MMP activation are vital for determining whether successful wound healing or chronic non-healing is observed[51]. In normal wound healing, MMP9 expression increases concurrent with fibroblast migration into the wound area, and MMP3 levels increase at re-epithelialization[51]. In this study, the levels of MMP9 secretion increased in *Lepr$^{db/+}$*-derived adipose tissue at 2 DPW, and MMP3 levels increased at 8 DPW. However, in *Lepr$^{db/d}$*-derived adipose tissue, MMP3 increased at both 2 DPW and 8 DPW, suggesting that dysregulation of MMPs in adipose tissue results in wound healing impairment in *Lepr$^{db/db}$* mice. Furthermore, the secretion levels of IGFBP3 and Serpin F1 in *Lepr$^{db/+}$* adipose tissue increased at 8 DPW. IGFBP3 is one of 6 structurally related IFGBPs that bind to IGF peptides with high affinity, and the IGFBP3•IGF1 complex binds to fibrin clots and concentrates at wound sites to facilitate wound healing[41]. Serpin F1 is a pigment epithelium-derived factor (PEDF), and PEDFs contribute to the resolution of wound healing by causing regression of immature blood vessels and stimulating maturation of the vascular micro-environment, which promotes a return to tissue homeostasis after injury[52]. Our findings suggest that *Lepr$^{db/+}$* adipose tissue–secreted SASP factors promote cell migration. By contrast, *Lepr$^{db/db}$* adipose tissue–secreted SASP factors may potentially induce wound healing impairment.

In this study, we focused on acute wounding in our mouse model. We have previously demonstrated the importance of transient senescence in the healing of acute muscle injury. Moreover, we found that this transient senescence does not occur during chronic muscle inflammation, which can be ameliorated by inducing senescence among mesenchymal stroma cells[53,54]. Therefore, we suspect that diabetes may cause changes to the characteristics of acute senescence post wounds on the skin. Following acute wounding of diabetic mice, senescence is delayed in subcutaneous adipose tissue; however, senescent cells are present in subcutaneous adipose tissue of diabetic patients over 60 d post-wounding, which suggests a chronic wound. Previous studies have demonstrated the importance of transiently induced senescence for acute wound healing whereas chronic senescence induces pathological conditions[55]. For example, transient senescence promotes collagen deposition after acute damage and limits fibrosis[8,10]. By contrast, SA-β-Gal+ senescent cells are present in chronic wounds, and the presence of more than 15% senescence in populations of cells isolated from venous ulcers may indicate a "difficult to heal" ulcer[16].

In this study, we could not determine how delayed senescence contributes to the accumulation of senescent cells in diabetic chronic wounds. While senescent cells generated by acute inflammation are removed by immune cells through the action of SASP, accumulated senescent cells acquire immune evasion and anti-apoptotic properties in the chronic inflammatory environment[53,54,56,57]. In the future, it will be necessary to determine how the senescence induction after cell damage is related to the accumulation of senescent cells found in diabetic wounds. Regarding the accumulation of senescent cells in subcutaneous adipose tissue, Rouault et al. recently reported that SA-β-Gal activity is seven times higher in subcutaneous than in omental adipose tissue when following a prospective cohort of

**Table1 Information of patients with diabetes and controls.**

| Donor/Group | Sex | Age | DM Daignosis | HbA1c (NGSP) [%] | Diagnosis | Wound (Source of the adipose tissue) | Time post-wounding | Comorbidity | Past Medical History(PMH) | Medication |
|---|---|---|---|---|---|---|---|---|---|---|
| Non-DM1 | Male | 65 | - | - | Electric injury, Burn | Abdomen | 19 | No | Postoperative hepatoma | Celecoxib, rebamipide |
| Non-DM2 | Male | 82 | - | - | Burn | Thigh | 39 | Chronic renal impairment, dementia | Myocardial infarction, cerebral infarction, postoperative colorectal cancer | Esomeprazole magnesium hydrate, acetaminophen, aspirin, carvedilol, amlodipine besilate, amiodarone hydrochloride, silodosin |
| Non-DM3 | Male | 65 | - | - | Burn | Lower leg | 39 | Hypertension, hypercholesterolemia, narcolepsy | - | Esomeprazole magnesium hydrate, loxoprofen sodium hydrate, amoxicillin hydrate, aspirin, carvedilol, clopidogrel sulfate, vitamin B6, vitamin B12 |
| Non-DM4 | Male | 45 | - | - | Burn | Hip | 39 | Smoke inhalation | Traumatic intracranial hemorrhage | Lactulose, meropenem |
| Non-DM5 | Female | 70 | - | - | Pressure ulcer | Chest | 70 | Chronic obstructive pulmonary disease(COPD) | Breast cancer | Fluticasone furoate, umeclidinium bromide vilanterol trifenatate, cilnidipine, allopurinol, sodium risedronate hydrate, rabeprazole sodium, domperidone, syakuyaku-kanzo-to(kampo medicine) |
| Non-DM6 | Male | 21 | - | - | Pressure ulcer | Ischial region | 84 | Spinal cord injury(Paraplegia) | - | - |
| DM1 | Female | 76 | + | 7.1 | Chronic traumatic ulcer | Elbow | 11 | Thrombocytopenic purpura, atrial fibrillation, iron deficiency anemia, dementia, gastric cancer | Thoracic compression fracture | Denosumab, prednisolone (15 mg/day), fenofibrate, pilsicainide hydrochloride hydrate, vonoprazan fumarate, clotiazepam, solifenacin succinate, sodium ferrous citrate, donepezil hydrochloride, magnesium oxide |
| DM2 | Male | 64 | + | 7.3 | Burn | Thigh | 13 | Hypertension | unknown | Rikkunshito (kampo medicine), omeprazole |
| DM3 | Male | 60 | + | 5.2 | Burn | Back | 17 | Multi-hepatoma, liver cirrhosis, esophageal varices, hypertension | - | Rikkunshito (kampo medicine), insulin human (genetical recombination), micafungin sodium, ceftazidime hydrate, linezolid, lactulose |
| DM4* | Male | 60 | + | 5.2 | Burn | Abdomen | 31 | Multi-hepatoma, liver cirrhosis, esophageal varices, hypertension | - | Acetaminophen, rikkunshito (kampo medicine), insulin human (genetical recombination), furosemide, ceftazidime hydrate, linezolid, lactulose, entecavir hydrate |

**Table 1 (continued)**

| Donor/Group | Sex | Age | DM Daignosis | HbA1c (NGSP) [%] | Diagnosis | Wound (Source of the adipose tissue) | Time post-wounding | Comorbidity | Past Medical History(PMH) | Medication |
|---|---|---|---|---|---|---|---|---|---|---|
| DM5 | Male | 70 | + | 11 | Fournier gangrene | Hip | 35 | Hypercholesterolemia | - | Canagliflozin hydrate, metformin hydrochloride, insulin degludec(genetical recombination), olmesartan medoxomil, pitavastatin calcium hydrate |
| DM6 | Female | 82 | + | 6.3 | Ulcer after vein harvesting | Lower leg | 64 | Gallstone, diabetic retinopathy, hypertension, hypercholesteremia | Postoperative abdominal aortic aneurysm, inflammation of mesenteric adipose tissue, ischemic enteritis, cholangitis aneurysm, interstitial lung disease | Acetaminophen, magnesium oxide, vildagliptin, sennoside A·B calcium, insulin lispro(genetical recombination), insulin glargine(genetical recombination), pilsicainide hydrochloride hydrate, aspirin, warfarin potassium, bisoprolol fumarate, vonoprazan fumarate, atorvastatin calcium hydrate |

*Same patient as DM3

227 individuals with severe obesity. Metabolic complications, including type 2 diabetes and dyslipidemia, are more prevalent in patients with high SA-β-Gal levels[58]. Moreover, eliminating accumulated senescent cells in adipose tissue by targeting them to the apoptotic pathway improved glucose tolerance, enhanced insulin sensitivity, lowered circulating inflammatory mediators, and promoted adipogenesis in obese mice[27]. Therefore, targeting the cellular senescence of subcutaneous adipose tissue may be effective for treating diabetic wounds.

We identified a transient increase in p15$^{INK4B}$ + senescent PDGFR-α + cells in adipose tissue during the normal wound healing process; however, in diabetic $Lepr^{db/db}$ mice, p15$^{INK4B}$ + senescent PDGFR-α + cells steadily accumulated over the course of wound healing. Accumulation of p15$^{INK4B}$ + senescent PDGFR-α + cells is also observed in diabetic patients, and non-diabetic patients have a decreased number of senescent cells during wound healing. PDGFR-α + mesenchymal stromal cells in skeletal muscle exhibit transient senescence, and this promotes muscle regeneration after acute muscle injury[53,54]. PDGFRα + or vimentin+ cells in neonatal hearts exhibit transient senescence to promote regeneration[6,7]. Using qRT-PCR and immunohistochemical analysis, we found that the levels of p15$^{INK4B}$, encoded by the $Cdkn2b$ gene, significantly increased in $Lepr^{db/+}$ adipose tissue at 2 DPW. CDKN2B expression levels are highest in subcutaneous adipose tissue relative to other tissues such as bone marrow, the central nervous system, skeletal muscle, and the internal organs[59]. Furthermore, CDKN2B expression levels in subcutaneous adipose tissue are correlated with BMI[59].

Although the detailed function of p15$^{INK4B}$ is not as well understood as those for p16$^{INK4A}$ and other senescence-related factors[60], the function and protein structure of p15$^{INK4B}$ are predicted to be similar to p16$^{INK4A}$, and p15$^{INK4B}$ is upregulated in TGF-β–related cell senescence[61]. p15$^{INK4B}$ binds to CDK4 and CDK6, preventing their binding to cyclins and thereby inhibiting cell cycle progression[61]. The antiproliferative action of TGF-β is also mediated through the inhibition of c-Myc expression. c-Myc inhibits the expression of p15$^{INK4B}$ and p21$^{CIP1/WAF1}$ in proliferating cells[62], and suppression of c-Myc by TGF-β limits c-Myc availability and suppresses the activity of p15$^{INK4B}$ and p21$^{CIP1/WAF1}$[63,64]. TGF-β expression and activation are rapidly induced in response to injury, and TGF-β controls wound healing by acting as a potent chemoattractant for monocytes and fibroblasts[65]. Although the role of TGF-β in the control of diabetic wound healing remains to be fully explained, increased levels of TGF-β1 are linked with type-2 diabetes, and TGF-β1 contributes to diabetic wound healing[66]. The regulation of TGF-β–related cell senescence may be a therapeutic target for diabetic ulcers.

Our results suggest that the transient increase in the number of senescent PDGFRα + cells in adipose tissue is important for wound healing, and that there is a decrease in cell senescence in diabetic wounds during the acute wound-healing phase. Hence, impairment of wound healing is concurrent with the accumulation of senescent cells. However, our study has several limitations. We analyzed subcutaneous adipose tissue; however, intradermal adipose tissue contributes to wound healing through regulation of adipocyte precursor proliferation and mature intradermal adipocyte repopulation in the skin after wounding[20]. The anatomical location of intradermal adipose tissue and subcutaneous adipose tissue is separated by the panniculus carnosus in mice, but human skin does not have a detectable panniculus carnosus. It is also difficult to confirm the existence of intradermal adipose tissue in humans because of the obvious inability to conduct lineage-tracing studies[67,68]. Even though intradermal and subcutaneous adipose tissue are not physically demarcated in humans, increasing evidence suggests that there is a functional distinction

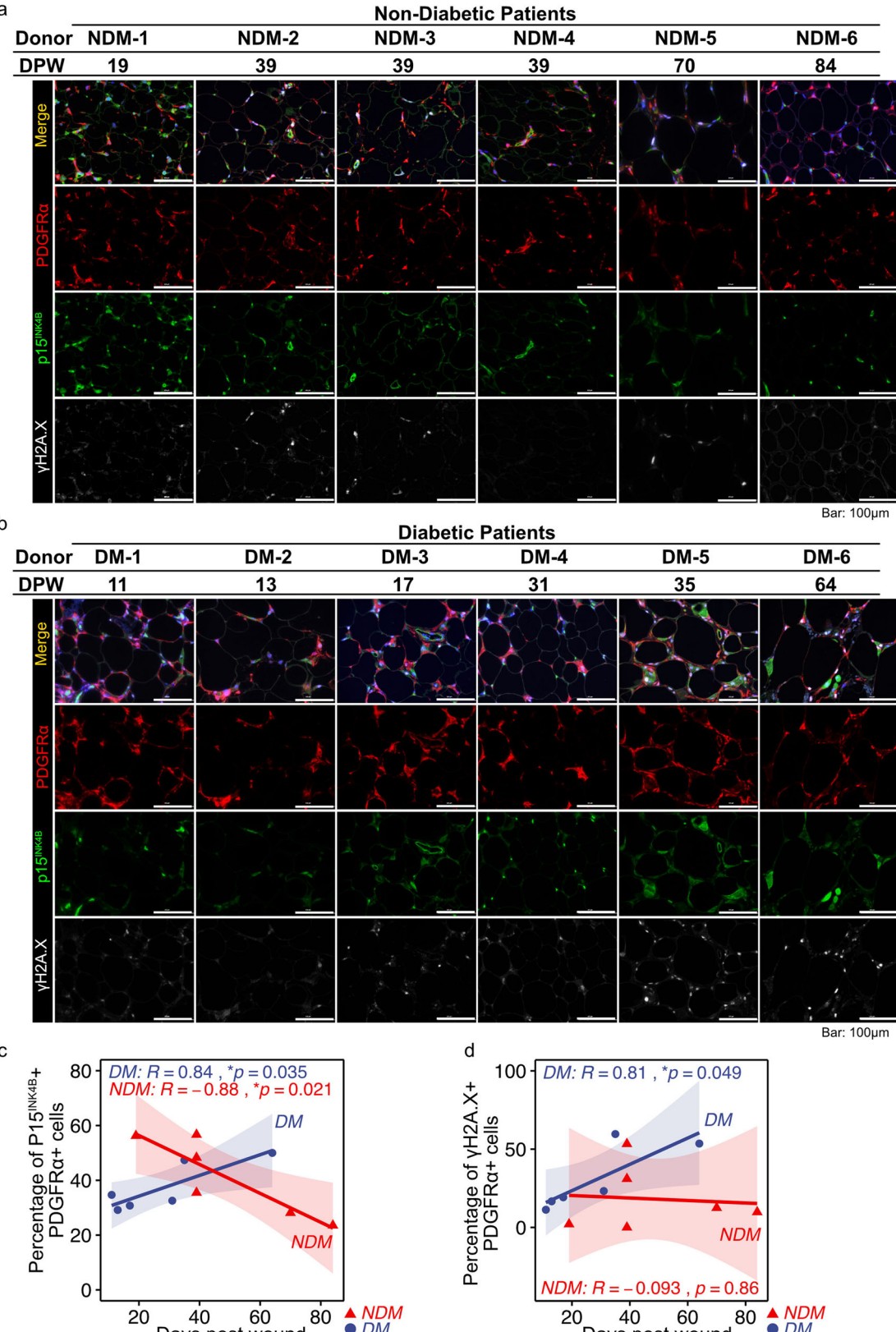

**Fig. 5 Cellular senescence in subcutaneous adipose tissue during wound healing in diabetic patients. a, b** Representative images of adipose tissue immunostained for PDGFRα, p15INK4B, and γH2A.X. during wound healing in diabetic and non-diabetic patients. **c, d** Correlation between the percentage of p15INK4B + or γH2A.X + cells in PDGFRα + cells and time post-wounding ($n = 6$ for each group). Correlations were examined statistically using Pearson's correlation coefficient, and 95% confidence intervals are shown with translucent fill corresponding to each marker color.

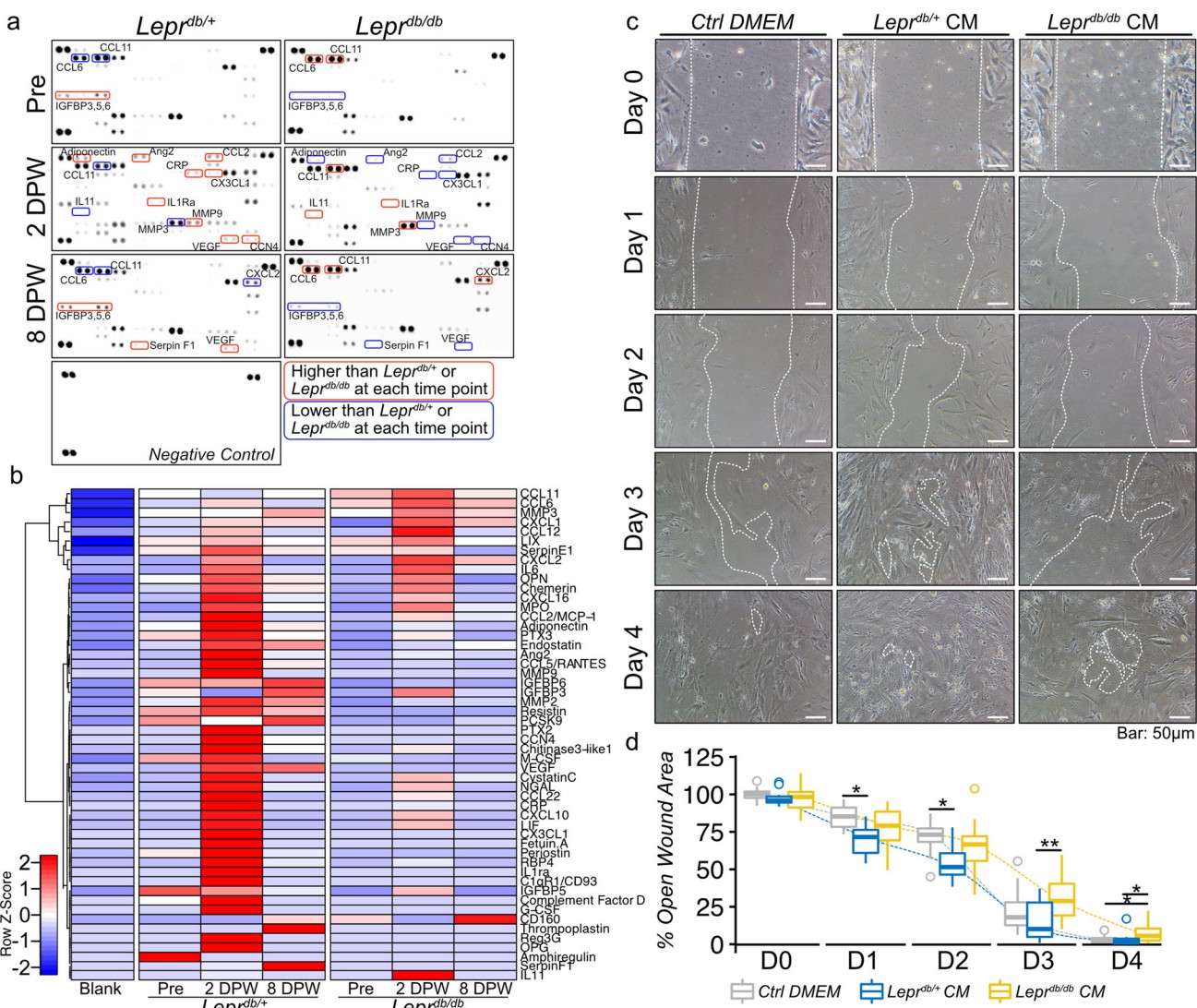

**Fig. 6 Components of post-wound adipose tissue-derived cytokines in *Lepr^db/+* and *Lepr^dbdb+* mice and their effect on cell migration. a** Proteome profiler antibody array analysis of SASP-containing culture media (CM) collected from the organ culture of adipose tissue at pre-wounding, 2 DPW, and 8 DPW. **b** Hierarchical clustering of differentially expressed cytokine expression. Cytokines with higher expression are depicted in red, cytokines with lower expression are depicted in blue, and cytokines with no difference are depicted in white. **c** Representative phase-contrast image of wound assay treated with adipose tissue-derived cultured media collected from either *Lepr^db/+* or *Lepr^db/db* mice at 2DPW and (**d**) the percentage of wound closure at each time point, from immediately after wounding (D0) to 4 d after wounding (D4) (Ctrl DMEM: *n* = 15; *Lepr^db/+* CM: *n* = 12; *Lepr^db/+* CM: *n* = 15). Quantitative data are shown as box-and-whisker plots with IQRs and 1.5 times the IQR. *p*-values were determined using the Tukey method for one-way ANOVA (*\*p* < 0.05 and *\*\*p* < 0.001).

between these tissues[68]. Hence, careful interpretation of our results is needed, but at a minimum, we have demonstrated that cellular senescence in subcutaneous adipose tissue contributes to both normal and diabetic wound healing. However, the interstitial connective tissue under the panniculus carnosus of mouse differs anatomically from that of humans. Mouse interstitial connective tissue separates the panniculus carnosus and subcutaneous adipose tissue. The muscular layer of the panniculus carnosus is located above interstitial connective tissue[35]. Although the muscular layer is mostly absent in humans, several muscles are considered as vestigial rudiments of the panniculus carnosus[35]. Recent studies demonstrated that human subcutaneous adipose tissue consists of fibrous connective tissue layer that demarcates two layers in subcutaneous adipose tissue: superficial and deep[69,70]. It is unclear which layer human stromal cells correspond to, but even in humans, myofibroblasts that may promote wound healing are regulated by subcutaneous adipose tissue.

Next, our study focused on PDGFRα + mesenchymal cell senescence in adipose tissue. Macrophages are among the senescent cell populations in diabetic wounds[15], and their abundance is correlated with senescent cell burden in adipose tissue[24–27]. Although PDGFRα + cells play important roles in adipose tissue homeostasis[71], further study is needed regarding the role that macrophage senescence plays in adipose tissue during wound healing. In this study, we used the *Lepr^db/db* mouse diabetes model, which is widely used as a typical delayed healing model[72], and our findings are similar to what is observed in type-2 diabetic patients. To better understand adipose tissue–related mechanisms in diabetic wound healing, further study is needed using a type-1 diabetes model, which exhibits adipose tissue atrophy and fibrosis[73–75]. Finally, we only used the male *Lepr^db/db* mouse diabetes model in this study, because several studies have shown that the prevalence of diabetes ulcers is higher among males than females[76,77] and that the risk of amputation in

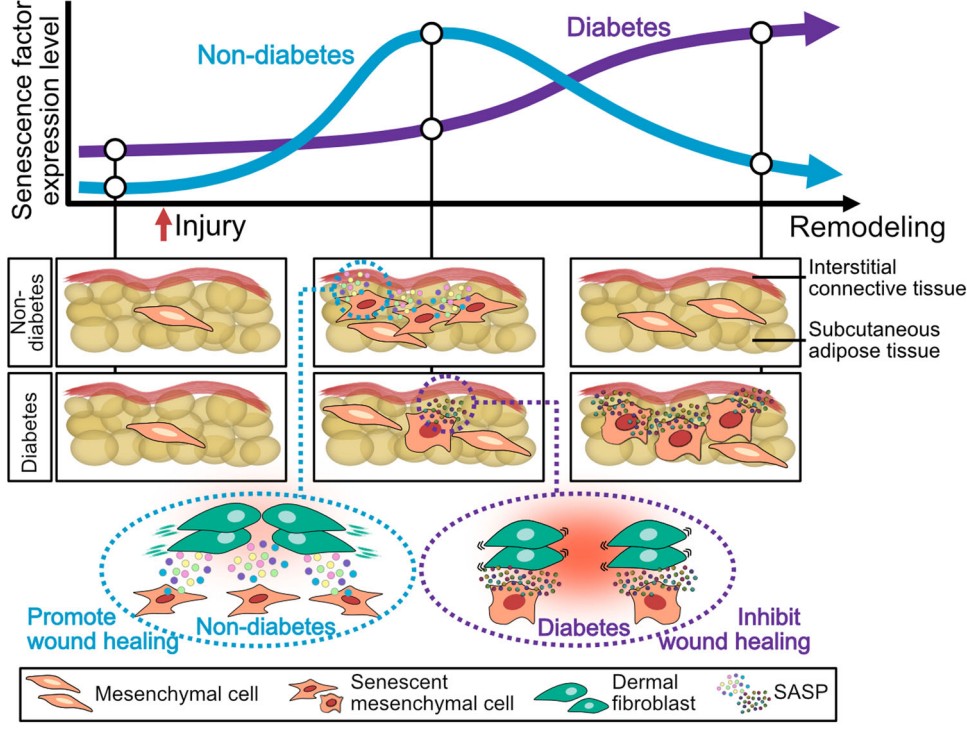

**Fig. 7 Schematic illustration of the contributions of senescent cells in adipose tissue during physiological and diabetic wound healing.** Transient mesenchymal cell senescence occurs in adipose tissue in physiological wound healing after injury; however, accumulation of cell senescence occurs concurrently with expression of different components of SASP in diabetic wound healing. Our data suggest that impairment of senescence in adipose tissue contributes to intractable wound healing in diabetes.

diabetic ulcer is higher in males than in females[78]. However, sex is acting as a modifier of wound healing mechanism[79]. Further studies are needed to elucidate common and sex-specific mechanisms of wound healing and cellular senescence. In summary, we demonstrate that transient mesenchymal cell senescence occurs in adipose tissue in physiological wound healing; however, accumulation of cell senescence occurs concurrently with expression of different components of the SASP in diabetic wound healing, suggesting that impairment of senescence in adipose tissue contributes to intractable wound healing in diabetes (Fig. 7).

## Methods

**Human wound samples**. Six wound tissue samples from five diabetic patients (age: 68.7 ± 9.0 years old; HbA1c(NGSP): 7.02 ± 2.15%) and six wound tissue samples from six non-diabetic patients (age: 58.0 ± 21.7 years old) were used in the study. The tissues used in the study were debrided and disposed during surgery for treatment, and no additional excisions were made for the study. The adipose tissue used in this study was included among the debrided tissue and was located in the peripheral wound site. Samples were recruited after agreement was obtained through informed consent. Detailed information about these subjects is given in Table 1. The Ethical Review Board at the Sapporo Medical University in Japan approved the study.

**Animals**. The Committee of the Animal Experimentation Center at the Sapporo Medical University School of Medicine approved all animal protocols. Mice were fed a standard chow diet and were kept on a 12 h light/12 h dark cycle with free access to food and water at all times. We used male C57BLKS/JIar-+Lepr$^{db}$/+Lepr$^{db}$ (Lepr$^{db/db}$), male C57BLKS/JIar-m + /+Lepr$^{db}$ (Lepr$^{db/+}$), and male C57BL/6 mice (Sankyo Lab Service, Tokyo, Japan) in the experiments. At 10–11 weeks of age, blood samples were taken from the tail to measure blood glucose, which was confirmed to be above 300 mg/dL in the diabetic mice and below 200 mg/dL in the control mice. Nipro Stat Strip XP2 (Nova Biomedical, Tokyo, Japan) was used for blood glucose level measurements.

**Wound model**. Mice were anesthetized using mixed anesthetic agents (medetomidine, midazolam, butorphanol)[80]. To generate the skin ulcer model, a 10 mm diameter, full-thickness circle excision was made on the backs of Lepr$^{db/db}$ and Lepr$^{db/+}$ mice. Wounds were photographed with a digital camera (COLPIX S9700; Nikon, Tokyo, Japan). Images were analyzed by tracing the wound margin, and the enclosed pixel area was calculated using the Image J software, version 1.5 (National Institutes of Health, Bethesda, MD, USA). The wound areas were standardized by measuring the captured image. Using the original wound size for comparison, the percentage of wound closure was calculated as follows: day n area/day 0 area × 100 (%).

**Adipose tissue transplantation**. To perform adipose tissue transplantation, 100–120 mg of subcutaneous adipose tissue was collected from the back of Lepr$^{db/db}$ and Lepr$^{db/+}$ mice and labeled with Cell Tracker Vybrant™ CM-DiI Cell-Labeling Solution (V22888; Thermo Fisher Scientific, MA, USA). Labeling of adipose tissue was performed according to the protocol. Briefly, tissue was incubated with CM-DiI at 37 °C for 5 min and at 4 °C for 15 min. The labeled adipose tissue was transplanted onto the back of Lepr$^{db/+}$ mice, and the wound was sutured with nylon thread. In Lepr$^{db/+}$ mice, sham surgery was performed without adipose tissue transfer.

**Histological analysis and wound healing scoring**. Wound tissue and adipose tissue were harvested with marginal skin and fixed in 4% paraformaldehyde at 4 °C overnight. The following day, the tissue was cut into 5 mm sections and was paraffin-embedded. Each slide was stained with H&E and Masson trichrome, which were used for histological wound-healing scoring. For scoring purposes[36], an examiner assessed re-epithelization, epithelial thickness index, keratinization, granulation tissue thickness, scar elevation index, and remodeling. Histology sections were randomized, blinded, and evaluated by three investigators. Each parameter was objectively quantified and scored using ImageJ software. In this validated scoring system, the lowest score, 0, is associated with an open unhealed wound as is evident immediately and within the first day post-injury, whereas the highest score, 12, is associated with a completely closed and healed wound without excessive scarring. All images were captured using a BZ-X700 fluorescence microscope (KEYENCE, Osaka, Japan).

**Immunofluorescence staining**. Paraffin-embedded sections were deparaffinized and rehydrated for immunostaining. Antigen retrieval was performed in a microwave oven (95–98 °C for 10 min) using citrate buffer (10 mM sodium citrate, pH 6.0). After cooling, the slides were washed twice with deionized water and once with 1X Tris-buffered saline with Tween-20 (TBST) for 5 min each. The sections were blocked with 1% bovine serum albumin (BSA) in TBST for 15 min at room temperature (RT) and were then incubated with primary antibodies overnight at

4 °C or for 1 h at RT. After washing three times with TBST for 5 min each, the sections were incubated with SignalStain Boost IHC Detection Reagent (HRP, Rabbit #8114; Cell Signaling Technology, Danvers, MA, USA) for 30 min at RT in the dark. The sections were then washed in TBST three times for 5 min each and treated with TSA Plus Working Solution (Fluorescein, Cyanine 3, and Cyanine 5; AKOYA BIOSCIENCES, Marlborough, MA, USA) for 10 min at RT in the dark. For multiplex staining, stripping was performed in a microwave oven (95–98 °C for 10 min) using citrate buffer. After cooling, staining with different tyramide fluorescent labels was performed according to the above procedure. Nuclei were labeled with Cellstain DAPI solution (1:1000, 4′,6-diamidino-2-phenylindole; Dojindo, Kumamoto, Japan), and after further washes, the sections were mounted in VECTASHIELD (Vector Laboratories, Burlingame, CA, USA). The following primary antibodies were used: rabbit anti-p15$^{INK4B}$ (1:500; ab53034; Abcam, Cambridge, UK), rabbit anti-p16$^{INK4A}$ (1:200; clone 5F22; Sigma-Aldrich), rabbit anti-p21 (1:1000; clone 5G7; Sigma-Aldrich), rabbit anti–PDGF receptor α (1:1000 (mouse) and 1:500 (human); D1E1E, XP; Cell Signaling Technology), rabbit anti–α-smooth muscle actin (1:500; D4K9N, XP; Cell Signaling Technology), and rabbit anti–phospho-histone H2A.X(Ser139) (1:480; Ser139, 20E3; Cell Signaling Technology). These primary antibodies were used after dilution with SignalStain Antibody Diluent (Cell Signaling Technology). All images were captured using a BZ-X700 fluorescence microscope (KEYENCE).

**SPiDER-β-Gal staining**. Gelatin-embedded frozen tissue was used. Ten micrometer-thick frozen tissue sections were washed three times with TBST and were incubated with SPiDER-β-Gal staining solution (1:50 in McIlvaine Buffer Solution, pH 6.0; Dojindo, Kumamoto, Japan) for 30 min at 37 °C. Nuclei were labeled with Cellstain DAPI solution (1:1000; Dojindo) and stained sections were mounted in VECTASHIELD (Vector Laboratories) after further washing.

**RNA extraction and quantitative reverse transcription PCR (qRT-PCR)**. Total RNA was isolated from skin wound tissues and subcutaneous adipose tissues using TRI Reagent (Molecular Research Center, Cincinnati, OH, USA) and was reverse transcribed into cDNA using the iScript cDNA Synthesis Kit (1708891; Bio-Rad). SsoAdvanced Universal SYBR Green Supermix (Bio-Rad) and a 7500 Real-Time PCR System (Applied Biosystems, Foster City, CA, USA) were used for qRT-PCR with the following conditions: 95 °C for 30 s and 40 amplification cycles of 95 °C for 15 s and 60 °C for 1 min. Expression levels were normalized to glyceraldehyde 3-phosphate dehydrogenase (GAPDH) or beta-actin (ACTB) levels. The primer sequences used for the PCR analysis are shown in Supplementary Table 1.

**Organ culture and SASP characterization**. Subcutaneous adipose tissue under the wound for five $Lepr^{db/+}$ or $Lepr^{db/db}$ mice in each group [0 DPW (before wound), 2 DPW, and 8 DPW] was collected and put into phosphate-buffered saline (PBS) supplemented with 2% penicillin/streptomycin. The tissue was then washed and transferred to Dulbecco's Modified Eagle's Medium (DMEM) supplemented with 10% fetal bovine serum (FBS). The volume of the culture medium was 1 mL for every 60 mg of adipose tissue, the medium was changed once at 24 h post-extraction, and the culture medium was collected at 48 h post-extraction for use in the following assay. The culture medium from adipose tissue culture was characterized using the Proteome Profiler Mouse XL Cytokine Array Kit (R&D Systems, Minneapolis, MN, USA). Developed films were imaged and integrated density analysis was performed using Image J, version 1.5 (National Institutes of Health).

**Cell preparation and in vitro wound-healing assays**. Skin fibroblasts were collected from male C57BL/6 mice. After euthanasia, skin was harvested and digested in Liberase TL (5401020001, Merck) for 120 min at 37 °C. The digested skin slurries were filtered through a 100 µm cell strainer (EASYstrainer Cell; Greiner Bio-One, Kremsmuenster, Austria) and through a 70 µm cell strainer (Greiner Bio-One). Cells were suspended in DMEM supplemented with 10% PBS and 1% penicillin/streptomycin and were cultured in a T75 culture flask. Cells reached 80–90% confluence after incubation for 1–2 weeks, and the cells were passaged. Cells from passage 2 were used for the in vitro studies. In vitro wound healing was studied using 2-well Culture-Inserts (Ibidi, Bavaria, Germany). Mouse skin-derived fibroblasts were cultured in 2-well Culture-Inserts with adipose tissue–cultured media collected from 2DPW adipose tissues. Phase-contrast images were obtained every 24 h and immediately after removing the 2-well Culture-Insert using Primovert and Axiocam208 microscopes (Carl Zeiss, Jena, Germany).

**Statistics and reproducibility**. Mice were only excluded from the study if they had visible wounds from fighting. The established scientific standard of $n \geq 3$ was applied throughout this study. All experiments were independently repeated using at least three mice per experimental group. We did not use any specific method of randomization to determine how animals were allocated to experimental groups. The investigators were not blinded to allocations during experiments or outcome assessments, although they were blinded during the histological assessment. Statistical analyses were performed using R (The R Foundation for Statistical Computing, Vienna, Austria). Statistical significance between two groups was determined using an unpaired t-test. A one-way or two-way analysis of variance

(ANOVA) was conducted to assess differences among three or more groups. Pairwise comparisons were made only when the ANOVA test identified a statistical significance. $p$-values for multiple comparisons were adjusted using the Tukey method. Statistical analyses were performed using EZR, which is a graphical user interface for R[81]. Two-sided $p$-values < 0.05 were considered statistically significant. Quantitative data are presented as either the mean ± standard error of the mean (SEM) or median with interquartile range (IQR) and 1.5 × IQR. Box-and-whisker plots and bar plots were generated using ggplot2, a plotting system for R based on The Grammar of Graphics (The R Foundation for Statistical Computing, Vienna, Austria). The R packages FactoMineR and factoextra were used to generate heat maps, Ward's hierarchical agglomerative clustering, and principal component analyses.

**Reporting summary**. Further information on research design is available in the Nature Research Reporting Summary linked to this article.

## Data availability
All data generated or analysed during this study are included in this published article and its supplementary information files. Source data can be found in Supplementary Data 1.

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

## Acknowledgements

We would like to thank Tatsuya Shiraishi, Naoko Sai, and Yumiko Takagi for technical support and ZENIS for providing native language editing services. This work was supported by JSPS KAKENHI (Grant Number 19K18908) and LEOC.

## Author contributions

A.K., Y.S., and T.S.C. designed the studies. A.K., Y.S., N.M., M.M, S.Y., T.S., and T.S.C. performed the experiments and analyzed the data. A.K., Y.S., Y.T., M.F., and T.S.C. contributed to the drafting and reviewing of the paper.

## Competing interests

The authors declare no competing interests.

## Additional information

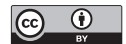

