## [Peer Review File · Communications Biology]

Reviewers' comments:

Reviewer #2 (Remarks to the Author):

Report

This study was performed to understand the role of adipose-derived mesenchymal cell senescence for wound healing. The study aimed to show that diabetic wounds in Lperdb/db mice exhibit low levels of senescent factors and accumulate senescent cells, impairing acute wound healing. Indeed, wound healing is a dynamic process. This study reported that p15INK4B+ senescent PDGFR- α + mesenchymal cells in adipose tissue are transiently increased during early wound healing in non-diabetic controls, promoting wound healing.

Transplantation of adipose tissue from diabetic mice to non-diabetic mice resulted in poor wound healing secondary to insufficient acute induction of senescence after injury. The study characterizes cellular senescence and SASP profiling from adipose tissue in diabetic and non-diabetic mice, and also shares data from diabetic patients.

The main limitations of this study are that senescence profiling was performed using p15INK4B marker, a newer marker of senescence compared to validated p16INK4A or p21. The overall senescence profiling therefore requires validation with known markers of senescence i.e., p16INK4A or p21. In addition, wound healing secondary to adipose tissue senescence was discussed without commentary on the role of skin senescence.

Diabetic wound healing is an important area of research as diabetic foot ulcers result in significant health care burden. This study therefore informs an unmet patient need.

Major Comments

Title: Consider including adipose tissue in the title to increase specificity: Altered regulation of mesenchymal cell senescence in adipose tissue promotes pathological changes associated with diabetic wound healing.

Figure 1: The study reported that Lperdb/db mice with higher body weight and blood glucose has poor cutaneous wound healing at 0 DPW (days post-wound) to 8 DPW by gross and histological evaluation. The study also suggests that diabetic mice exhibit a diffuse distribution of PDGFR- α and α -SMA expression at 8 DPW in the subcutaneous adipose tissue in contrast to non-diabetic mice which exhibit a localized distribution in the panniculus carnosus layer (present only in vertebrates) at the wound edge. The relevance for this variability in expression pattern is unclear as it relates to epidermal wound healing.

- Please include PDGFR- α and α -SMA immunostaining in the epidermal and dermal tissue at 8 DPW for diabetic and non-diabetic mice.

Figure 2: The study focuses on the role of subcutaneous adipose tissue in wound healing.

Therefore, the authors transplanted adipose tissue from diabetic mice into excisional skin of non-diabetic mice. It is reported that excisional skin wound was sutured closed after transplantation to stabilize the transplanted tissue. This allows for the excisional skin covering to act as a natural barrier (versus an open wound).

- Please include a control group with excisional skin closure suture closed with sham transplant (no transplanted tissue). This will evaluate the role of skin barrier alone without adipose tissue transplant.

- Figure 2C depicts a wound healing histological score which is arbitrary and nonblinded. It does not add value.

- Figure 2D shows 'normal dermis and wound region' but this is an overmagnified area. Please include wound region with epidermis and lower magnification image.

- Figure 2E could benefit from showing the percentage of CM-Dil labeled adipose tissue at 2 DPW and 8 DPW.

- To complement the immunostaining, please include RNA expression levels for PDGFR- α and α -SMA in adipose tissue at 8 DPW for diabetic and non-diabetic mice.

Figure 3: This study evaluated cellular senescence in adipose tissue from diabetic and nondiabetic mice using p15INK4B marker, which was acknowledged in the discussion as a newer marker of senescence compared to p16INK4A. Multiple senescence-related genes were analyzed (Cdkn2b, Cdkn1a, Trp53, Serpine1, Serpine2, IL-6 and TGF- β 1) in adipose tissue and compared to skin tissue (supplemental figure). While this paper focuses on the role of adipose tissue in wound healing, its relation to skin will be important to consider.

- Consider including skin mRNA data along with adipose mRNA data in Figure 3A in a side-by-side comparison.
- It was noted that p15INK4B, which is encoded by Cdkn2b, is transiently increased at 2 DPW in non-diabetic mice and this transient early expression contributes to normal wound healing. This also corresponds to Figure 3A mRNA expression of Cdkn2b in adipose tissue. The supplemental figure shows no change in expression for Cdkn2b in 2 DPW and 8 DPW skin between diabetic and non-diabetic mice. How do you explain the discrepancy in expression in adipose and skin tissues with this gene and other genes? The role of cell senescence during wound healing in skin is previously defined. There needs to be more discussion about the interplay between adipose tissue and skin regarding senescent markers or an explanation for the variability.
- To complement the immunostaining, please include RNA expression levels for PDGFR- α and p15INK4B in adipose tissue from diabetic and non-diabetic mice.
- Recommend senescence β -Galactosidase staining for adipose tissue from diabetic and non-diabetic mice.
- Highly recommend validation with p16INK4A and/or p21 senescence markers.

Figure 4: This study reports cellular senescence in adipose tissue from diabetic and nondiabetic patients. Representative images are shown from immunostaining of adipose tissue however the source of the adipose tissue is unclear. The patients are relatively age-matched between diabetic and non-diabetic patients however the source of wound significantly varies from burn injury versus pressure ulcer versus chronic traumatic ulcer according to the table. There are also variances in days post-wound (less than 30 days – acute wound and greater than 30 days – chronic wound). It is very interesting that the percentage of senescence markers increases in diabetic patients with days post-wound, especially with patient 6 with diabetes mellitus who is an elderly patient with lower extremity ulcer. Therefore, the potential differences in acute wound healing (shown in this paper) versus chronic wound healing could be highlighted in the discussion section, especially to suggest that this paper focuses on the acute wound model only.

- Is the adipose tissue from patients collected from the peripheral wound site or an arbitrary non-wound site (i.e., abdomen)? Please clarify.
- Highly recommend validation with p16INK4A and/or p21 senescence markers.

Figure 5: SASP profiling was performed to assess cytokines released from adipose tissue in diabetic and non-diabetic mice collected from under the wound bed.

- Clarify the source for adipose-derived conditioned media for Figure 5C (i.e., pre-, 2 DPW, or 8 DPW).

Minor comments

The study was conducted objectively and under ethical guidelines. There could be areas for improvement in methodology details as previously suggested. There are proper controls and defined outcomes.

Line 65: Reference 15 (senescence and the healing rates of venous ulcers) evaluates venous ulcers not diabetic ulcers. Please update to venous ulcers if still relevant to include here.

Line 282: The study comments on the induction of chronic inflammation however the results of the study span an acute timeframe (8 DPW) therefore, recommend adjusting wording to acute inflammation.

Reviewer #3 (Remarks to the Author):

This is an interesting and insightful study provided by Kita et al that adds to the current limited literature on cellular senescence and wound repair. Overall a number of interesting experiments have been performed to elucidate the role of subcutaneous adipose tissue in repair, and its links to cellular senescence. However, the authors have oversensationalised their data in areas, and this should be addressed. Specific comments below.

Introduction

Line 52: "Increased senescent cells are observed in the human skin during normal wound healing, heart regeneration in zebrafish and neonatal mice, and tissue regeneration in salamanders and zebrafish 4–7". There are no references here that links senescence to human skin healing. Please amend the text to reflect this.

Paragraph beginning line 58 is a bit difficult to understand as there is no clear definition of how senescence in normal and chronic healing differs. This should be amended to clearly explain that senescence in normal wound healing is transient (i.e. induced to allow matrix formation and then regresses) and in chronic wounds is uncontrolled and sustained.

Results

Figure 1/Line 106-111: It would be more informative to provide quantification of collagen and adipose tissue for Fig 1d rather than just making observational differences with no quantitative data to confirm this. Please measure the adipose tissue area within the wound and provide quantification of collagen from the Masson's Trichrome staining.

Line 118-120: It is difficult to determine from the images alone whether there are differences in α -SMA and PDGFR- α +ve cells between the non-diabetic and diabetic mice. Please provide quantification of these cells to strengthen the provided results. Arrows pointing the +ve cells would also guide the reader further. Was the focus only on the subcutaneous region beneath the panniculus carnosus?

Line 127: It would be more informative to provide a less vague measurement of wound healing than histological score which is based on observation alone. Wound width/wound area/re-epithelialisation/granulation tissue maturity would give more concrete evidence. Please provide one or more of these additional analyses to strengthen the data.

Line 133: If stating that a group has increased levels of α -SMA+ve and PDGFR- α +ve cells in a particular region then quantification of the cells should also be shown, not just representative images. Please provide this analysis.

Line 188: How might the delay in adipose tissue senescence in diabetic mice allow for accumulation of senescent cells in diabetic mice? Please move this statement to the discussion and justify it there.

Line 229/ Fig5c-d: I find it hard to believe that the fibroblast scratches have taken 6 days to close? The measurement as % Closure is also opposite to what it should be and may cause confusion. There isn't 100% closure at D0. Please analyse in the opposite direction or alternatively describe this as % open scratch area.

Discussion

Line 237: Until quantification is performed on senescent cell numbers from immunohistochemical analysis then it cannot be assumed that there is an increase in senescent cells (see results comments).

Line 251: Please specify that this is in vitro wound healing. "which affects wound healing in vitro"

Line 278: It is a stretch to say that the findings of the study suggest Lepr db/+ SASP promote wound healing by inducing cell migration and proliferation, and inhibit fibrosis as the only

functional readout performed was the scratch assay. Please rephrase this to state cell migration only, and that further work would be needed to assess the influence of these adipose tissue derived SASP factors on proliferation/fibrosis.

Line 281: Again, it is an overinterpretation to state the Lepr db/db SASP induce chronic inflammation as this has not been assessed, either in vitro or looked at in the transplantation model. Please rephrase this to be more speculative... the adipose-tissue SASP may potentially induce chronic inflammation...

Line 284: Quantitative data is needed to be able to make the claim that a transient increase in senescence is observed in the mice (see comments on results section).

Methods

Line 253 onwards – animal use. It needs to be made clear that the male mice were used for the wounding study and the female mice for fibroblast isolation for in vitro analysis (in the animal section and the figure legends). Why were only male mice used in the wounding study, and why was the opposite sex used for in vitro study? Please clarify the groupings further and reasoning for this. The limitation of using one sex should also be added to the discussion. N numbers need to be included and justified.

We thank the Reviewers for their careful consideration of our manuscript. We have revised the manuscript by taking into account each point raised by the reviewers. These changes are highlighted in “red” color fonts in the revised manuscript. We have now addressed each of the comments as outlined below.

Reviewers' comments:

Reviewer #2 (Remarks to the Author):

Report

This study was performed to understand the role of adipose-derived mesenchymal cell senescence for wound healing. The study aimed to show that diabetic wounds in *Lper^{db/db}* mice exhibit low levels of senescent factors and accumulate senescent cells, impairing acute wound healing. Indeed, wound healing is a dynamic process. This study reported that p15^{Ink4b+} senescent PDGFR- α + mesenchymal cells in adipose tissue are transiently increased during early wound healing in non-diabetic controls, promoting wound healing.

Transplantation of adipose tissue from diabetic mice to non-diabetic mice resulted in poor wound healing secondary to insufficient acute induction of senescence after injury. The study characterizes cellular senescence and SASP profiling from adipose tissue in diabetic and non-diabetic mice, and also shares data from diabetic patients.

Author Response: We thank Reviewer #2 for the positive comments and suggestions.

The main limitations of this study are that senescence profiling was performed using p15^{Ink4b} marker, a newer marker of senescence compared to validated p16^{Ink4a} or p21. The overall senescence profiling therefore requires validation with known markers of senescence i.e., p16^{Ink4a} or p21. In addition, wound healing secondary to adipose tissue senescence was discussed without commentary on the role of skin senescence.

Diabetic wound healing is an important area of research as diabetic foot ulcers result in significant health care burden. This study therefore informs an unmet patient need.

Author Response: Thank you for your comments. We performed IHC analysis with antibodies against p21^{CIP1/WAF1} and p16^{INK4A} and found that 75% of p15^{INK4B}-positive cells expressed either p16^{INK4A}, p21^{CIP1/WAF1}, or both in mice. Moreover, 80% of p15^{INK4B}-positive cells expressed p16^{INK4A}, p21^{CIP1/WAF1}, or both in humans. As described in the Discussion (page 18, lines 366–373), p15^{INK4B}, p16^{INK4A}, and p21^{CIP1/WAF1} share a common feature, namely senescence. p15^{INK4B} is similar to p16^{INK4A} in that both bind to CDK4 and CDK6, thereby inhibiting cell-cycle progression. In addition, p15^{INK4B} and p21^{CIP1/WAF1} have been shown to be associated with TGF- β mediated cellular senescence. Although these senescence markers were not completely merged, at least over 75% of p15^{INK4B}-positive cells are senescent cells supported by another senescence marker. Please see the Results section (pages 10–11, lines 207–210), Supplementary Fig. 2a–b (pages 10-11, lines 207-210), and Supplementary Fig. 4a-c (page 12, lines 231–234).

We also described the discrepancy of senescence role in the skin and adipose tissue in the Discussion. Please see pages 14, lines 277–290, and author response #2–8.

Major Comments

Title: Consider including adipose tissue in the title to increase specificity: Altered regulation of mesenchymal cell senescence in adipose tissue promotes pathological changes associated with diabetic wound healing.

Author Response: Thank you for your suggestion. We agree with the reviewer and have changed the title to “Altered regulation of mesenchymal cell senescence in adipose tissue promotes pathological changes associated with diabetic wound healing.”

Figure 1: The study reported that *Lper^{db/db}* mice with higher body weight and blood glucose has poor cutaneous wound healing at 0 DPW (days post-wound) to 8 DPW by gross and histological evaluation. The study also suggests that diabetic mice exhibit a diffuse distribution of PDGFR- α and α -SMA expression at 8 DPW in the subcutaneous adipose tissue in contrast to non-diabetic mice which exhibit a localized distribution in the panniculus carnosus layer (present only in vertebrates) at the wound edge. The relevance for this variability in expression pattern is unclear as it relates to epidermal wound healing.

#2-1

- Please include PDGFR- α and α -SMA immunostaining in the epidermal and dermal tissue at 8 DPW for diabetic and non-diabetic mice.

Author Response: We have added low-magnification images of both epidermal and dermal tissue. Please see Fig. 1h.

#2-2

Figure 2: The study focuses on the role of subcutaneous adipose tissue in wound healing. Therefore, the authors transplanted adipose tissue from diabetic mice into excisional skin of non-diabetic mice. It is reported that excisional skin wound was sutured closed after transplantation to stabilize the transplanted tissue. This allows for the excisional skin covering to act as a natural barrier (versus an open wound).

- Please include a control group with excisional skin closure suture closed with sham transplant (no transplanted tissue). This will evaluate the role of skin barrier alone without adipose tissue transplant.

Author Response: In response, we have added histological data of the sham group that is an excisional skin closure suture closed with sham transplant. The sham group showed earlier wound healing than mice transplanted with *Lper^{db/db}* adipose tissue (*Lper^{db/db}* ATT) but no difference from mice transplanted with *Lper^{db/+}* (*Lper^{db/+}* ATT). Please see Fig. 2a–c, and the Results (page 7, lines 142–

144).

#2-3

- Figure 2C depicts a wound healing histological score which is arbitrary and nonblinded. It does not add value.

Author Response: Recently, Vyver et al. reported that a validated histology scoring system based on the different parameters in each phase of healing can be easily and accurately assessed using either hematoxylin and eosin (H&E) or Masson's Trichrome (MT) staining¹. Three investigators assigned the histological score in a blinded fashion. The intraclass correlation coefficient (ICC) is 0.79, which indicates a good degree of agreement². We added new wound healing histological score, and transplantation of adipose tissue derived from *Lep^{db/db}* mice into *Lep^{db/+}* mice significantly impaired wound healing. Please see the Results (page 7, lines 140–142) and Fig. 2c.

#2-4

- Figure 2D shows 'normal dermis and wound region' but this is an overmagnified area. Please include wound region with epidermis and lower magnification image.

Author Response: We have added low-magnification images of epidermal and dermal tissue. Please see Fig. 2d.

#2-5

- Figure 2E could benefit from showing the percentage of CM-Dil labeled adipose tissue at 2 DPW and 8 DPW.

Author Response: We have added the percentage of CM-Dil positive cells at 2 DPW and 8 DPW. Please see Fig. 2h.

#2-6

- To complement the immunostaining, please include RNA expression levels for PDGFR- α and α -SMA in adipose tissue at 8 DPW for diabetic and non-diabetic mice.

Author Response: Thank you for your comments. In this figure, we focused on the change of dermal tissue to understand whether wound healing is impaired or not by transplanted adipose tissue. Therefore, we performed PCR analysis in skin tissue to complement the immunostaining data. The mRNA expression of *Acta2*, which is a gene of α -SMA, was not significantly different, but *Pdgfra* was significantly higher in the skin tissue of *Lep^{db/db}* ATT mice compared with *Lep^{db/+}* ATT mice. Please see Fig. 2f and the Results (page 8, lines 149–151).

Figure 3: This study evaluated cellular senescence in adipose tissue from diabetic and nondiabetic mice using p15^{Ink4b} marker, which was acknowledged in the discussion as a newer marker of senescence compared to p16^{Ink4a}. Multiple senescence-related genes were analyzed (Cdkn2b, Cdkn1a, Trp53, Serpine1, Serpine2, IL-6 and TGF-β1) in adipose tissue and compared to skin tissue (supplemental figure). While this paper focuses on the role of adipose tissue in wound healing, its relation to skin will be important to consider.

#2-7

- Consider including skin mRNA data along with adipose mRNA data in Figure 3A in a side-by-side comparison.

Author Response: Thank you for your suggestion. We have put the skin mRNA data along with adipose tissue mRNA data in Fig. 3 to allow for side-by-side comparison. We also added additional data of *Il6* and *Tgfb1* expression in skin to compare with skin and adipose tissue. We also added the principal component analysis of skin in Fig. 3b. Please see Fig. 3a–c.

#2-8

- It was noted that p15^{Ink4b}, which is encoded by *Cdkn2b*, is transiently increased at 2 DPW in non-diabetic mice and this transient early expression contributes to normal wound healing. This also corresponds to Figure 3A mRNA expression of *Cdkn2b* in adipose tissue. The supplemental figure shows no change in expression for *Cdkn2b* in 2 DPW and 8 DPW skin between diabetic and non-diabetic mice. How do you explain the discrepancy in expression in adipose and skin tissues with this gene and other genes? The role of cell senescence during wound healing in skin is previously defined. There needs to be more discussion about the interplay between adipose tissue and skin regarding senescent markers or an explanation for the variability.

Author Response: Given that the expression pattern of senescence-related genes, including *Cdkn2b*, determined by PCA did not differ between *Lepr^{db/db}* and *Lepr^{db/+}* in wounded skin, skin senescence should happen in diabetic wounds in a similar manner to normal wound healing. On the other hand, in the subcutaneous adipose tissue, we observed a rapid and transient increase in senescent cells in control; however, the number of senescent cells in diabetic *Lepr^{db/db}* adipose tissue gradually increased post-wounding. Transplantation of subcutaneous adipose tissue from diabetic *Lepr^{db/+}* mice into normal dermal tissue caused deteriorated wound healing and PDGFRα-positive cells increased in *Lepr^{db/db}* interstitial connective tissue layer, while α-SMA positive myofibroblasts of interstitial connective tissue infiltrated to the wound site with adipose tissue in control. These data indicate that the senescence in subcutaneous adipose tissue may induce differentiation of myofibroblasts from PDGFRα-positive cells of the control interstitial connective tissue layer, but the delay of cellular senescence and difference of the SASP components in diabetes subcutaneous adipose tissue may limit differentiation to myofibroblasts from PDGFRα-positive cells, resulting in impairment of wound

healing. We have added these descriptions in the Discussion (pages 13–14, lines 277–290).

#2-9

- To complement the immunostaining, please include RNA expression levels for PDGFR- α and p15^{Ink4b} in adipose tissue from diabetic and non-diabetic mice.

Author Response: Thank you for your suggestion. The level of *Cdkn2b* is now shown in Fig. 3a; *Cdkn2b* encodes p15^{INK4B}. We have added the level of *Pdgfra* mRNA and data showing that *Pdgfra* gene expression increased at 8 DPW in *Lepr^{db/db}* mice ($p = 0.065$), although this increase is not statistically significant. Please see Fig. 4c, and the Results (page 10, lines 199–201).

#2-10

- Recommend senescence β -Galactosidase staining for adipose tissue from diabetic and non-diabetic mice.

Author Response: Thank you for your comments. We have added images and quantitative data concerning SPiDER- β -Gal, which is a newly developed fluorescence-based assay wherein activation of fluorescence is observed upon reaction with SA- β -galactosidase(Gal)³. We found that cells positive for SPiDER- β -Gal fluorescence increased in *Lepr^{db/+}* non-diabetic mice at 2 DPW but decreased at 8 DPW, while SPiDER- β -Gal-positive cells increased in *Lepr^{db/db}* diabetic mice only at 8 DPW, which agree with our p15^{INK4B} immunostaining results. Please see Supplemental Fig. 3 and the Results (page 11, lines 211–215).

#2-11

- Highly recommend validation with p16^{Ink4a} and/or p21 senescence markers.

Author Response: Thank you for your comments. We performed IHC analysis with p21^{CIP1/WAF1} and p16^{INK4A} antibodies. The results showed that 75% of p15^{INK4B}-positive cells expressed either p16^{INK4A}, p21^{CIP1/WAF1}, or both in mice. Please see Supplemental Fig. 2 and the Results (pages 10–11, lines 207–210).

#2-12

Figure 4: This study reports cellular senescence in adipose tissue from diabetic and nondiabetic patients. Representative images are shown from immunostaining of adipose tissue however the source of the adipose tissue is unclear. The patients are relatively age matched between diabetic and non-diabetic patients however the source of wound significantly varies from burn injury versus pressure ulcer versus chronic traumatic ulcer according to the table. There are also variances in days post-wound (less than 30 days – acute wound and greater than 30 days – chronic wound). It is very interesting that the percentage of senescence markers increases in diabetic patients with days post-wound, especially with patient 6 with diabetes mellitus who is an elderly patient with lower extremity ulcer.

Therefore, the potential differences in acute wound healing (shown in this paper) versus chronic wound healing could be highlighted in the discussion section, especially to suggest that this paper focuses on the acute wound model only.

Author Response: Thank you for your comments. We have added the source of the adipose tissue in Table 1. We have also added Past Medical History. Please see Table 1. In this study, we focused on acute wounding in the mice model. We previously demonstrated the importance of transient senescence in the healing process of acute muscle injury; such transient senescence is not observed in chronic muscle inflammation, which can be ameliorated by inducing senescence of mesenchymal stroma cells^{4,5}. Therefore, we suspected that diabetes may induce changes in the characteristics of acute senescence post-wounding on the skin. During acute wounding of diabetic mice, senescence induction was delayed in subcutaneous adipose tissue, while senescent cells were present in the subcutaneous adipose tissue of diabetic patients over 60 days post-wounding that resembles chronic wounding.

Previous studies have demonstrated the importance of transiently induced senescence for acute wound healing, while chronic senescence induces pathological conditions⁶. For example, transient senescence promotes collagen deposition after acute damage and limits fibrosis. By contrast, SA- β -Gal-positive senescent cells are present in chronic wounds, and when more than 15% of cells isolated from venous ulcers are senescent, this may indicate a “difficult to heal” ulcer⁷. In this study, we could not clear how delayed senescence contributes to the accumulation of senescent cells in diabetic chronic wounds. While senescent cells generated by acute inflammation are removed by immune cells through the action of SASP, accumulated senescent cells become capable of immune evasion and resistant to apoptosis in the chronic inflammatory environment^{4,5,8,9}. In the future, it will be necessary to investigate how the speed of senescence induction after cell damage correlates with the accumulation of senescent cells found in diabetic wounds. We have added the above descriptions to the Discussion (page 16, lines 322–351).

#2-13

- Is the adipose tissue from patients collected from the peripheral wound site or an arbitrary non-wound site (i.e., abdomen)? Please clarify.

Author Response: All patient-derived adipose tissues used in this study were obtained from either the marginal area or from directly under the wound. We have added this information to the Methods (page 21, lines 432–433).

#2-14

- Highly recommend validation with p16^{Ink4a} and/or p21 senescence markers.

Author Response: Thank you for your comments. We performed IHC analysis with p21^{CIP1/WAF1} and p16^{INK4A} antibodies. The results showed that 80% of p15^{INK4B} positive cells expressed either p16^{INK4A},

p21^{CIP1/WAF1}, or both in humans. Please see Supplemental Fig. 4 and the Results (page 12, line 231–234).

#2-15

Figure 5: SASP profiling was performed to assess cytokines released from adipose tissue in diabetic and non-diabetic mice collected from under the wound bed.

- Clarify the source for adipose-derived conditioned media for Figure 5C (i.e., pre-, 2DPW, or 8 DPW).

Author Response: We have added details concerning the sources of adipose-derived conditioned media in the Results (page 13, line 261–263) and the legend for Fig. 6.

Minor comments

#2-16

The study was conducted objectively and under ethical guidelines. There could be areas for improvement in methodology details as previously suggested. There are proper controls and defined outcomes.

Author Response: We have modified details of our methodology, including the sources of adipose tissue in the human experiment, the sources of adipose-derived conditioned media, and added descriptions of the controls for the mouse experiment.

#2-17

Line 65: Reference 15 (senescence and the healing rates of venous ulcers) evaluates venous ulcers not diabetic ulcers. Please update to venous ulcers if still relevant to include here.

Author Response: Thank you for pointing out this error. We have changed “diabetic ulcers” to “venous ulcers” as requested. Please see the Introduction (page 4, lines 67–68).

#2-18

Line 282: The study comments on the induction of chronic inflammation however the results of the study span an acute timeframe (8 DPW) therefore, recommend adjusting wording to acute inflammation.

Author Response: Thank you for your comment. We have changed the indicated sentence to “By contrast, *Lep^r^{db/db}* adipose tissue–secreted SASP factors may potentially induce wound healing impairment.” Please see page 16, lines 319–321.

Reviewer #3 (Remarks to the Author):

This is an interesting and insightful study provided by Kita et al that adds to the current limited literature on cellular senescence and wound repair. Overall a number of interesting experiments have been performed to elucidate the role of subcutaneous adipose tissue in repair, and its links to cellular senescence. However, the authors have oversensationalised their data in areas, and this should be addressed. Specific comments below.

Author Response: We thank Reviewer #3 for the comments and suggestions.

Introduction

#3-1

Line 52: “Increased senescent cells are observed in the human skin during normal wound healing, heart regeneration in zebrafish and neonatal mice, and tissue regeneration in salamanders and zebrafish 4–7”. There are no references here that links senescence to human skin healing. Please amend the text to reflect this.

Author Response: We apologize for the misdescription. We have changed the human skin to mouse skin and added a corresponding reference. Please see the Introduction (page 3, lines 52–54).

#3-2

Paragraph beginning line 58 is a bit difficult to understand as there is no clear definition of how senescence in normal and chronic healing differs. This should be amended to clearly explain that senescence in normal wound healing is transient (i.e. induced to allow matrix formation and then regresses) and in chronic wounds is uncontrolled and sustained.

Author Response: We have modified the description. Please see the Introduction (pages 3–4, lines 63–66).

Results

#3-3

Figure 1/Line 106-111: It would be more informative to provide quantification of collagen and adipose tissue for Fig 1d rather than just making observational differences with no quantitative data to confirm this. Please measure the adipose tissue area within the wound and provide quantification of collagen from the Masson’s Trichrome staining.

Author Response: We added quantification data of collagen and adipose tissue. Please see Fig. 1f–g, and the Results (page 6, lines 112–113).

#3-4

Line 118-120: It is difficult to determine from the images alone whether there are differences in α -SMA and PDGFR- α positive cells between the non-diabetic and diabetic mice. Please provide quantification of these cells to strengthen the provided the results. Arrows pointing the +ve cells would also guide the reader further. Was the focus only on the subcutaneous region beneath the panniculus carnosus?

Author Response: We added quantitative data of the percentage of PDGFR α and α -SMA-positive cells in interstitial connective tissue and granulation tissue. The data showed the percentage of α -SMA-positive cells decreased and the percentage of PDGFR α - and α -SMA-double positive cells increased in *Lepr^{db/db}* mice than that of *Lepr^{db/+}* mice, which indicates that *Lepr^{db/db}* mice exhibit diffuse distribution of α -SMA- and PDGFR α -positive cells on the interstitial connective tissue and granulation tissue at 8 DPW. Please see Fig. 1i-j, Results section pages 6–7, lines 118–132. We also added a new supplemental figure, which includes arrows that clearly indicate cells positive for PDGFR α and α -SMA. Please see Supplemental Fig. 1.

We focused on the interstitial connective tissue and subcutaneous adipose tissue beneath panniculus carnosus, because we observed the infiltration of myofibroblasts into both the interstitial connective tissue layer and adipose tissue at the acute wound site as shown in Fig. 1.

We have also added the anatomical definition of the panniculus carnosus and interstitial connective tissue to the Results and the Discussion. Please see the Results, page 7, line 127–129, and the Discussion, page 19–20, line 395–402.

#3-5

Line 127: It would be more informative to provide a less vague measurement of wound healing than histological score which is based on observation alone. Wound width/wound area/re-epithelialisation/granulation tissue maturity would give more concrete evidence. Please provide one or more of these additional analyses to strengthen the data.

Author Response: Recently, Vyver et al. reported that a validated histology scoring system based on the different parameters in each phase of healing can be easily and accurately assessed using either hematoxylin and eosin (H&E) or Masson's trichrome (MT) staining¹. This histological score including 1) percentage of re-epithelization, 2) epithelial thickness, 3) keratinization, 4) granulation tissue thickness, 5) scar elevation index, which is an indirect assessment of scar formation based on the presence of a hypertrophied dermis in the wound area, and 6) remodeling score that is a semi-quantitative assessment dependent on the presence of any or all of dermal white adipose tissue, skin appendage, and collagen deposition¹. In this study, three investigators each assigned a histological score in a blinded fashion. The intraclass correlation coefficient (ICC) is 0.79, which indicates good reliability². We added the wound healing histological score, and transplantation of adipose tissue derived from *Lepr^{db/db}* mice into *Lepr^{db/+}* mice significantly impaired wound healing. Please see the Results (page 7, lines 140–142) and Fig. 2c.

#3-6

Line 133: If stating that a group has increased levels of α -SMA⁺ and PDGFR- α ⁺ cells in a particular region then quantification of the cells should also be shown, not just representative images. Please provide this analysis.

Author Response: We have added quantitative data, namely the percentage of PDGFR α -positive cells, α -SMA-positive cells, and cells positive for both PDGFR α and α -SMA in interstitial connective tissue. We added these quantitative data to determine if *Lepr*^{db/db} mice-derived adipose tissue transplantation induces or impairs wound healing compared with normal and diabetic wound healing as seen in Fig. 1h–1j. We found that mice transplanted with *Lepr*^{db/db}-derived adipose tissue (*Lepr*^{db/db} ATT) showed an increased population of cells positive for both PDGFR α and α -SMA in interstitial connective tissue, which was also observed for impaired wound healing in *Lepr*^{db/db} mice. Please see Fig. 2d–e, Results section, pages 7–8, lines 145–149.

#3-7

Line 188: How might the delay in adipose tissue senescence in diabetic mice allow for accumulation of senescent cells in diabetic mice? Please move this statement to the discussion and justify it there.

Author Response: We could not determine how delayed senescence contributes to the accumulation of senescent cells in diabetic chronic wounds in this study. We moved the statement to the Discussion as requested and discussed the accumulation of senescent cells in adipose tissue in diabetic mice as follows.

“In this study, we could not determine how delayed senescence contributes to the accumulation of senescent cells in diabetic chronic wounds. While senescent cells generated by acute inflammation are removed by immune cells through the action of SASP, accumulated senescent cells acquire immune evasion and anti-apoptotic properties in the chronic inflammatory environment^{4,5,8,9}. In the future, it will be necessary to determine how the senescence induction after cell damage is related to the accumulation of senescent cells found in diabetic wounds. Regarding the accumulation of senescent cells in subcutaneous adipose tissue, Rouault et al. recently reported that SA- β -Gal activity is seven times higher in subcutaneous than in omental adipose tissue when following a prospective cohort of 227 individuals with severe obesity. Metabolic complications, including type 2 diabetes and dyslipidemia, are more prevalent in patients with high SA- β -Gal levels¹⁰. Moreover, eliminating accumulated senescent cells in adipose tissue by targeting them to the apoptotic pathway improved glucose tolerance, enhanced insulin sensitivity, lowered circulating inflammatory mediators, and promoted adipogenesis in obese mice²⁷. Therefore, targeting the cellular senescence of subcutaneous adipose tissue may be effective for treating diabetic wounds.”

Please see Discussion section, pages 17, lines 336–351.

#3-8

Line 229/ Fig5c-d: I find it hard to believe that the fibroblast scratches have taken 6 days to close? The measurement as % Closure is also opposite to what it should be and may cause confusion. There isn't 100% closure at D0. Please analyse in the opposite direction or alternatively describe this as % open scratch area.

Author Response: We performed additional experiment using C57BL/6 male mice skin-derived fibroblast, and compared the wound healing among three groups: control group, which cultured with DMEM; *Lepr^{db/+}* adipose tissue-derived conditioned media; and *Lepr^{db/db}* adipose tissue-derived conditioned media. The control condition closed the wound after 4 d, and *Lepr^{db/db}* adipose tissue-derived conditioned media delayed wound closure. Conversely, *Lepr^{db/+}* adipose tissue-derived conditioned media accelerated wound closure after 1 and 2 d compared with the control.

Regarding the time taken for wound closure, several studies have reported that it takes around 48 h, but other studies reported that about 20–50% of wounds close within 36–48 h after scratching a mouse fibroblast monolayer^{12–16}, which agrees with our results. Another reason for our experiment took long time, we used 2-well Culture-Inserts that are available for purchase from Ibdid□. These two-well silicone inserts enable reproducible generation of highly defined gaps without devoid of cells by removing them after cells have achieved confluency. This experiment system did not cause the stimulation seen in conventional scratch assays that involve scratching a confluent cell layer with pipet tips. This may be why wound closure takes four days.

Overall, we used culture inserts to perform our wound healing assay, and added control conditions. Hence, we think our data reliably shows that adipose tissue-derived conditioned media affects fibroblast migration.

To prevent misunderstanding of our assay protocol, we changed the term “scratch assay” to “wound healing assay.”

Furthermore, we have fixed the label of the vertical axis to “% open wound area.”

Please see Fig. 6c–d, and page 13, lines 261–265.

Discussion

#3-9

Line 237: Until quantification is performed on senescent cell numbers from immunohistochemical analysis then it cannot be assumed that there is an increase in senescent cells (see results comments).

Author Response: Thank you for your comments. We showed the quantitative data of percentage of p15^{INK4B} positive in PDGFR α + cells and fluorescence intensity of p15^{INK4B} PDGFR α + cells. Please see Fig. 4 and the Results section (page 10, lines 199–207). In addition, we added new quantitative data of the percentage of SPiDER- β -Gal+ cells, which is a newly developed fluorescence-based assay wherein activation of fluorescence is observed upon reaction with SA- β -Gal³. Please see Supplemental Fig. 3 and the Results section (page 11, lines 211–215). From these results, we write that “we observed a rapid but transient increase in senescent cells in subcutaneous adipose tissue for the control; however, the number of senescent cells in diabetic *Lepr^{db/db}* adipose tissue gradually increased post-wounding”

#3-10

Line 251: Please specify that this is in vitro wound healing. “which affects wound healing in vitro”

Author Response: We have fixed it. Please see the revised manuscript, page 14, line 292.

#3-11

Line 278: It is a stretch to say that the findings of the study suggest *Lepr^{db/+}* SASP promote wound healing by inducing cell migration and proliferation, and inhibit fibrosis as the only functional readout performed was the scratch assay. Please rephrase this to state cell migration only, and that further work would be needed to assess the influence of these adipose tissue derived SASP factors on proliferation/fibrosis.

Author Response: We have rephrased this statement into “*Lepr^{db/+}* adipose tissue–secreted SASP factors promote cell migration.” Please see the Discussion section, page 16, lines 318–319.

#3-12

Line 281: Again, it is an overinterpretation to state the *Lepr^{db/db}* SASP induce chronic inflammation as this has not been assessed, either in vitro or looked at in the transplantation model. Please rephrase this to be more speculative... the adipose-tissue SASP may potentially induce chronic inflammation...

Author Response: Reviewer #2 also suggested that this statement is overinterpretation. Therefore, we have rephrased this sentence into “By contrast, *Lepr^{db/db}* adipose tissue–secreted SASP factors may potentially induce wound healing impairment.” Please see the Discussion, page 16, lines 319–321.

#3-13

Line 284: Quantitative data is needed to be able to make the claim that a transient increase in senescence is observed in the mice (see comments on results section).

Author Response: As we responded to your Comment #9, we added the quantitative data of the percentage of p15^{INK4B} positive in PDGFR α + cells, fluorescence intensity of p15^{INK4B} PDGFR α + cells, and the percentage of SPiDER- β -Gal+ cells. Please see Fig. 4 and the Result section (page 10–11, lines 201–210) and Supplemental Fig. 3, the Results section (page 11, lines 211–215).

Methods

#3-14

Line 253 onwards – animal use. It needs to be made clear that the male mice were used for the wounding study and the female mice for fibroblast isolation for in vitro analysis (in the animal section and the figure legends). Why were only male mice used in the wounding study, and why was the opposite sex used for in vitro study? Please clarify the groupings further and reasoning for this. The limitation of using one sex should also be added to the discussion. N numbers need to be included and justified.

Author Response: Thank you for your point out. In this study, we used female mouse-derived skin fibroblast previously developed to reduce the number of animals used. However, as you suggested, sex difference is a very important point. Several studies showed that the prevalence of diabetes ulcers was higher among males than females¹⁷. Furthermore, The risk of amputation is a sequela of diabetic foot ulceration, which causes huge morbidity and mortality, is higher in males than females¹⁸. Due to the high risk of diabetes ulcers, therefore we used male *Lepr^{db/db}* mice in this study. Now we added a new result of wound healing assay using C57BL/6 male mice skin-derived fibroblasts to show the effect of diabetic and non-diabetic adipose tissue-derived secretome factors on cell migration in vitro. This study used only male mice in an in vivo study, which we acknowledge is a limitation of this study. Hence, we added the limitation statement in the Discussion section. Please see Fig. 6c–d, the Results section (page 13, lines 258–265) and the Discussion (pages 20–21, lines 414–419). We also provide the N numbers in the legend of Figure 6.

1. van de Vyver, M. *et al.* Histology Scoring System for Murine Cutaneous Wounds. *Stem Cells Dev.* (2021) doi:10.1089/scd.2021.0124.
2. Koo, T. K. & Li, M. Y. A Guideline of Selecting and Reporting Intraclass Correlation Coefficients for Reliability Research. *J. Chiropr. Med.* **15**, 155–163 (2016).
3. Doura, T. *et al.* Detection of LacZ-Positive Cells in Living Tissue with Single-Cell Resolution. *Angew. Chemie - Int. Ed.* **55**, 9620–9624 (2016).
4. Chikenji, T. S. *et al.* p16INK4A-expressing mesenchymal stromal cells restore the senescence–clearance–regeneration sequence that is impaired in chronic muscle inflammation. *EBioMedicine* **44**, 86–97 (2019).
5. Saito, Y., Chikenji, T. S., Matsumura, T., Nakano, M. & Fujimiya, M. Exercise enhances skeletal muscle regeneration by promoting senescence in fibro-adipogenic progenitors. *Nat. Commun.* **11**, 889 (2020).
6. Wilkinson, H. N. & Hardman, M. J. Senescence in Wound Repair: Emerging Strategies to Target Chronic Healing Wounds. *Front. Cell Dev. Biol.* **8**, 1–13 (2020).
7. Stanley, A. & Osler, T. Senescence and the healing rates of venous ulcers. *J. Vasc. Surg.* **33**, 1206–1211 (2001).
8. Zhu, Y. *et al.* The Achilles' heel of senescent cells: from transcriptome to senolytic drugs. *Aging Cell* **14**, 644–658 (2015).
9. Ovadya, Y. *et al.* Impaired immune surveillance accelerates accumulation of senescent cells and aging. *Nat. Commun.* **9**, 5435 (2018).
10. Rouault, C. *et al.* Senescence-associated β -galactosidase in subcutaneous adipose tissue associates with altered glycaemic status and truncal fat in severe obesity. *Diabetologia* **64**, 240–254 (2021).
11. Palmer, A. K. *et al.* Targeting senescent cells alleviates obesity- induced metabolic dysfunction. *Aging Cell* **18**, e12950 (2019).
12. Long, M. *et al.* DPP-4 inhibitors improve diabetic wound healing via direct and indirect promotion of epithelial-mesenchymal transition and reduction of scarring. *Diabetes* **67**, 518–531 (2018).
13. Kung, H. N., Yang, M. J., Chang, C. F., Chau, Y. P. & Lu, K. S. In vitro and in vivo wound

- healing-promoting activities of β -lapachone. *Am. J. Physiol. - Cell Physiol.* **295**, C931–C943 (2008).
14. Lee, J.-Y., Oh, N. & Park, K.-S. Ell3 Modulates the Wound Healing Activity of Conditioned Medium of Adipose-derived Stem Cells. *Dev. Reprod.* **21**, 335–342 (2017).
 15. Oh, E. J. *et al.* Extracellular vesicles derived from fibroblasts promote wound healing by optimizing fibroblast and endothelial cellular functions. *Stem Cells* **39**, 266–279 (2021).
 16. Liu, X. *et al.* Direct comparison of the potency of human mesenchymal stem cells derived from amnion tissue, bone marrow and adipose tissue at inducing dermal fibroblast responses to cutaneous wounds. *Int. J. Mol. Med.* **31**, 407–415 (2013).
 17. Jalilian, M., Sarbarzeh, P. A. & Oubari, S. Factors related to severity of diabetic foot ulcer: A systematic review. *Diabetes, Metab. Syndr. Obes. Targets Ther.* **13**, 1835–1842 (2020).
 18. Fan, L. & Wu, X. J. Sex difference for the risk of amputation in diabetic patients: A systematic review and meta-analysis. *PLoS One* **16**, e0243797 (2021).

Reviewers' comments:

Reviewer #2 (Remarks to the Author):

I appreciate the authors for addressing reviewer comments. Regarding the comment on senescence profiling with p15ink4b marker, the authors provided IHC analysis with antibodies against p21 and p16. The supplemental data was reviewed. Supplemental Figure 2 was not convincing for 75% p15ink4b positive cells expressing either p16, p21 or both. For Supplemental Figure 2, please show higher magnification. The current images are not convincing for 75% single or dual positivity with p16 and/or p21. How were these mouse p16 and p21 antibodies validated? It seems that the p16 and p21 antibodies work better for human diabetic patients as shown in Supplemental Figure 4. Regarding comment 2-12, you may consider adding a model figure that depicts how senescence in adipose tissue plays a role in wound healing. The remainder of comments were adequately addressed.

Reviewer #3 (Remarks to the Author):

The authors have provided a well revised manuscript that has addressed my queries relating to the study, including adding more detailed quantification and clarification of data. I have no further comments and support the manuscript being accepted for publication.

We thank the Reviewers for their careful consideration of our manuscript. These changes are highlighted in “red” color fonts in the revised manuscript. We have now addressed each of the comments as outlined below.

Reviewers' comments:

Reviewer #2 (Remarks to the Author):

I appreciate the authors for addressing reviewer comments. Regarding the comment on senescence profiling with p15ink4b marker, the authors provided IHC analysis with antibodies against p21 and p16. The supplemental data was reviewed. Supplemental Figure 2 was not convincing for 75% p15ink4b positive cells expressing either p16, p21 or both. For Supplemental Figure 2, please show higher magnification. The current images are not convincing for 75% single or dual positivity with p16 and/or p21. How were these mouse p16 and p21 antibodies validated? It seems that the p16 and p21 antibodies work better for human diabetic patients as shown in Supplemental Figure 4. Regarding comment 2-12, you may consider adding a model figure that depicts how senescence in adipose tissue plays a role in wound healing. The remainder of comments were adequately addressed.

Author Response: We appreciate Reviewer #2 for the comments and suggestions. We have added high-magnification images of Supplemental Figure 2. Regarding the p16 and p21 antibodies, we used ZooMAb recombinant antibodies. Each antibody is validated to reliably perform across multiple applications including IHC

(<https://www.sigmaaldrich.com/JP/en/products/protein-biology/antibodies/zoomab-antibodies>). Both anti-p16-INK4a antibody (clone 5F22 antibody) and anti-p21 antibody (clone 5G7) are rabbit monoclonal antibodies and species reactivities are human and mouse. Please see the website

<https://www.sigmaaldrich.com/JP/en/product/sigma/zrb1437> (for anti-p16 antibody)

<https://www.sigmaaldrich.com/JP/en/product/sigma/zrb1141> (for anti-p21 antibody).

We also added a model figure that depicts how senescence in adipose tissue plays a role in wound healing. Please see Figure 7.

Reviewer #3 (Remarks to the Author):

The authors have provided a well revised manuscript that has addressed my queries relating to the study, including adding more detailed quantification and clarification of data. I have no further comments and support the manuscript being accepted for publication.

Author Response: We thank Reviewer #3 for the positive comments.

REVIEWERS' COMMENTS:

Reviewer #2 (Remarks to the Author):

The authors have provided the requested additional information and answered the questions related to the study. The additional illustration (Figure 7) of the wound healing and senescence model is excellent and clearly articulates the proposed concept. No further comments. I fully support the manuscript for acceptance to publication.